# Kuramoto Orientation Diffusion Models

**Yue Song[1], T. Anderson Keller[2], Sevan Brodjian[1], Takeru Miyato[3,4],
Yisong Yue[1], Pietro Perona[1], and Max Welling[4,5]**

[1] Caltech    [2] Harvard University    [3] University of Tübingen
[4] University of Amsterdam    [5] CuspAI

## Abstract

Orientation-rich images, such as fingerprints and textures, often exhibit coherent angular directional patterns that are challenging to model using standard generative approaches based on isotropic Euclidean diffusion. Motivated by the role of phase synchronization in biological systems, we propose a score-based generative model built on periodic domains by leveraging stochastic Kuramoto dynamics in the diffusion process. In neural and physical systems, Kuramoto models capture synchronization phenomena across coupled oscillators – a behavior that we re-purpose here as an inductive bias for structured image generation. In our framework, the forward process performs *synchronization* among phase variables through globally or locally coupled oscillator interactions and attraction to a global reference phase, gradually collapsing the data into a low-entropy von Mises distribution. The reverse process then performs *desynchronization*, generating diverse patterns by reversing the dynamics with a learned score function. This approach enables structured destruction during forward diffusion and a hierarchical generation process that progressively refines global coherence into fine-scale details. We implement wrapped Gaussian transition kernels and periodicity-aware networks to account for the circular geometry. Our method achieves competitive results on general image benchmarks and significantly improves generation quality on orientation-dense datasets like fingerprints and textures. Ultimately, this work demonstrates the promise of biologically inspired synchronization dynamics as structured priors in generative modeling. Code is available at:https://github.com/KingJamesSong/OrientationDiffusion.

## 1   Introduction

Synchronization phenomena, characterized by coordinated rhythmic activity across coupled oscillators, are ubiquitous in nature. Such patterns are fundamental in biological neural networks, where synchronous neural firing supports critical cognitive functions including attention, sensory integration, and memory [9]. Similarly, in physical and engineering systems, synchronization underpins the collective behavior of coupled oscillatory circuits, chemical reactions, and mechanical structures [36, 62]. Central to understanding these phenomena is the Kuramoto model [36], a canonical framework arising from nonlinear dynamics that elegantly describes how global coherence emerges spontaneously from interactions among oscillatory units.

In this work, we explore how these principles of synchronized coherence can be harnessed to tackle a persistent challenge in generative modeling for *orientation-rich data*, such as fingerprints, textures, and directional fields. These data appear in numerous applications, including fingerprint generation for biometric security [17], material characterization through crystal orientation analysis [43], and fiber orientation modeling for improved medical diagnostics [47, 64]. The defining structures of these data types are characterized primarily by the orientations of local features (*i.e.,* piece-wise constant

39th Conference on Neural Information Processing Systems (NeurIPS 2025).

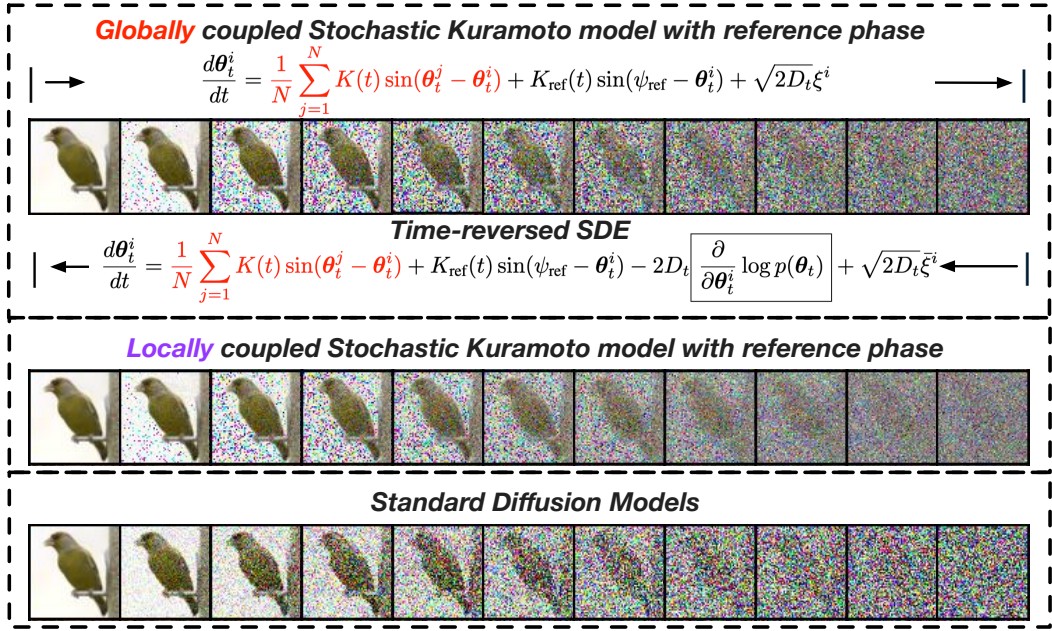

Figure 1: Governing stochastic differential equations (SDEs) and representative image samples from our globally and locally coupled Kuramoto orientation diffusion models. Pixels are mapped onto periodic domains as angular phase variables. In the globally coupled model, each pixel interacts with all other pixels via Kuramoto sinusoidal coupling (highlighted in red). The locally coupled variant corresponds to a similar SDE but restricts this sinusoidal coupling to a local neighborhood around each pixel. Unlike standard diffusion models, our approach introduces non-isotropic noise dynamics via pulling similar phases together, enabling a more structured destruction process. These dynamics help preserve the global structure in the early stages of diffusion (*e.g.,* the overall shape of the bird), while allowing for faster adaptation to noise as the process progresses. The forward SDE **synchronizes** phase variables through oscillator interactions and a global reference phase. The reverse process **desynchronizes** these variables using learned score functions to synthesize images.

signals with sharp transitions) rather than by raw pixel intensities. Crucially, such orientation patterns exist on periodic domains where angular discontinuities are problematic for conventional models that do not explicitly account for periodicity. Early attempts to handle these issues date back to the seminal work on orientation diffusions for image denoising [48], which highlighted how treating angular data without considering its circular nature can lead to artifacts and loss of coherence. Those findings underscore the need for specialized generative frameworks that explicitly account for the periodic structure inherent in directional data.

In this paper, we introduce a novel nonlinear score-based generative framework that leverages stochastic Kuramoto dynamics to operate directly on periodic domains. Synchronization offers a powerful inductive bias here: it encourages local patterns and orientations to reinforce one another – edges to align, ridges to remain coherent, and flows to stay smooth – before noise erodes these features. In our framework, pixel values are first mapped to angular phase variables, which enables natural compatibility with circular geometry and allows the diffusion process to evolve through interactions among phase coupling. We present our governing SDEs and illustrative examples of diffusion dynamics in Fig. 1. Our forward diffusion process performs *structured destruction* by progressively synchronizing angular phase variables through phase coupling and attraction to a common reference phase, driving the data distribution toward a low-entropy von Mises state. The Kuramoto interactions induce non-isotropic dynamics by pulling similar phases together, helping preserve local orientation and making the model particularly suitable for orientation-dense images.

The reverse generative process then performs desynchronization, leveraging learned periodic score functions to gradually reintroduce variability and reconstruct the image. The generation thus follows a hierarchical process, where the global structure is established first, and finer details are subsequently introduced, following a coarse-to-fine paradigm. We also propose a locally coupled variant of the

model that aligns with the spatial correlations of image data. Due to the coherence imposed by synchronization, our model converges faster to the terminal distribution during the forward process, which in turn enables the reverse process to generate high-quality samples within fewer diffusion steps. To handle periodicity robustly, we employ periodicity-aware neural networks with sinusoidal embeddings and define forward transitions using wrapped Gaussian kernels. The score function is then estimated by sampling from these local transitions derived from the forward dynamics.

Experiments on orientation-rich datasets, such as fingerprints, textures, and terrains, demonstrate that our Kuramoto orientation diffusion model consistently produces higher-fidelity images compared to standard diffusion baselines, often requiring fewer diffusion steps. Furthermore, our method remains competitive even on general CIFAR-10 benchmarks. Beyond images, we also report results on Earth/climate datasets on the 2D sphere and on Navier–Stokes fluid velocity fields, where the synchronization prior aligns with natural periodicity and angular structure, delivering consistent gains. Overall, this work bridges neural oscillation theory and modern score-based generative models, underscoring the potential of biologically inspired synchronization dynamics as structured priors.

## 2 Kuramoto Orientation Diffusion Models

### 2.1 Preliminary: Score-based Generative Models

In score-based generative models [3, 60], the forward and reverse processes of diffusion models are formulated as a pair of coupled SDEs:

$$
\begin{aligned}
&\texttt{Forward-SDE: } d\boldsymbol{x} = \boldsymbol{f}(\boldsymbol{x}, t)\, dt + g(t)\, d\boldsymbol{w} \\
&\texttt{Reverse-SDE: } d\boldsymbol{x} = \left[\boldsymbol{f}(\boldsymbol{x}, t) - g^2(t)\, \boxed{\nabla_{\boldsymbol{x}} \log p(\boldsymbol{x})}\, \right] dt + g(t)\, d\bar{\boldsymbol{w}}
\end{aligned}
\tag{1}
$$

where $f(\boldsymbol{x}, t)$ represents the vector-valued drift function, $g(t)$ denotes the scalar function of the diffusion coefficient, and $\boldsymbol{w}$ and $\bar{\boldsymbol{w}}$ are the standard Wiener processes. The boxed term represents the score function, which corresponds to the gradient of the log-density. A neural network $s(\boldsymbol{x}_t, t)$ is trained to approximate the score function by minimizing the following objective:

$$
\mathcal{L} = \mathbb{E}_{t \sim U(1,T)} \mathbb{E}_{\boldsymbol{x}_0} \mathbb{E}_{\boldsymbol{x}_t \sim p(\boldsymbol{x}_t|\boldsymbol{x}_0)} \left[ g^2(t) \left\| s(\boldsymbol{x}_t, t) - \nabla_{\boldsymbol{x}_t} \log p(\boldsymbol{x}_t|\boldsymbol{x}_0) \right\|_2^2 \right]
\tag{2}
$$

Common choices for the forward SDE include the Variance-Preserving (VP) and Variance-Exploding (VE) formulations [59, 60, 23]. Typically, the drift function is chosen to be linear, ensuring that the conditional distribution $p(\boldsymbol{x}_t|\boldsymbol{x}_0)$ remains analytically Gaussian and the corresponding score function $\nabla_{\boldsymbol{x}_t} \log p(\boldsymbol{x}_t|\boldsymbol{x}_0)$ can be computed in closed form.

### 2.2 Stochastic Kuramoto Models with Reference Phase

Fig. 2 illustrates the forward and reverse processes in our Kuramoto orientation diffusion model. During the forward process (left-to-right), we progressively destroy image information through synchronization, modeled by the following stochastic Kuramoto dynamics [36]:

$$
\frac{d\boldsymbol{\theta}_t^i}{dt} = \frac{1}{N} \sum_{j=1}^{N} K(t) \sin(\boldsymbol{\theta}_t^j - \boldsymbol{\theta}_t^i) + K_{\text{ref}}(t) \sin(\psi_{\text{ref}} - \boldsymbol{\theta}_t^i) + \sqrt{2D_t}\xi^i
\tag{3}
$$

where each colored circle represents an oscillator with periodic phase $\boldsymbol{\theta}_t^i \in [-\pi, \pi]$, $K(t)$ is the time-varying coupling strength among oscillators, $K_{\text{ref}}(t)$ is the coupling strength to a reference phase $\psi_{\text{ref}}$, $2D_t$ denotes the variance of Gaussian noise, and $N$ denotes the number of oscillators. The white rectangle represents the global reference phase $\psi_{\text{ref}}$, serving as an attractor to guide the population of oscillators toward a synchronized target. To measure the level of synchronization, we compute the complex ordering parameter:

$$
r(t)e^{i\psi(t)} = \frac{1}{N} \sum_{j=1}^{N} e^{i\boldsymbol{\theta}_t^j}
\tag{4}
$$

where $\psi(t) \in [-\pi, \pi]$ denotes the mean phase, and $r(t) \in [0, 1]$ measures the coherence of these phases. They correspond to the angle and length of the arrow in Fig. 2, respectively. When $r = 0$,

the system is completely *desynchronized*. When $r = 1$, the oscillators are *perfectly synchronized*. For intermediate values $0 < r < 1$, the system exhibits *partial synchronization*, where phases are clustered around the mean phase $\psi(t)$ with dispersion induced by noise.

**Phase Wrapping.** After each Kuramoto update step, we apply phase wrapping to ensure that all phase variables remain within the interval $[-\pi, \pi]$. Specifically, the wrapping is performed as $\boldsymbol{\theta} = (\boldsymbol{\theta} + \pi) \bmod (2\pi) - \pi$ where $\bmod$ denotes the modulo operation. This guarantees that the periodicity of the phase variables is consistently maintained throughout the forward process.

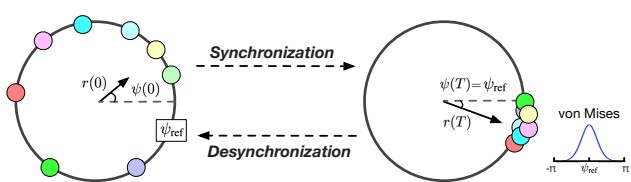

Figure 2: Illustration of our Kuramoto orientation diffusion model. In the forward process (left-to-right), angular phase variables (colored circles) synchronize toward a low-entropy von Mises distribution, guided by attraction to a reference phase (white rectangle). The reverse process (right-to-left) uses learned score functions to desynchronize these phases.

**Quasi-equilibrium.** In the thermodynamic limit ($N \to \infty$), the collective dynamics of the oscillator population can be effectively described by the mean-field evolution of a single representative oscillator with phase $\theta$. The mean-field dynamics of Eq. (3) follow the Fokker–Planck equation:

$$\frac{\partial p(\theta, t)}{\partial t} = -\frac{\partial}{\partial \theta}\left[\left(K(t)r(t)\sin(\psi(t) - \theta) + K_{\text{ref}}(t)\sin(\psi_{\text{ref}} - \theta)\right)p(\theta, t)\right] + D_t \frac{\partial^2 p(\theta, t)}{\partial \theta^2} \quad (5)$$

In our framework, akin to standard diffusion models, the coupling strengths and noise variance gradually increase over time. Consequently, the system does not reach a true stationary distribution. Instead, after an initial transient phase, the evolution of $p(\theta, t)$ becomes sufficiently slow such that, at each moment, the system can be approximated well by an instantaneous steady state – a regime we refer to as *quasi-equilibrium*, characterized by $\partial p_{\text{st}}(\theta)/\partial t \approx 0$.

Under quasi-equilibrium, the phase distribution approximately satisfies:

$$p_{\text{st}}(\theta) \approx \frac{1}{Z} \exp\left(\frac{K(T)r(T)}{D_T}\cos(\psi(T) - \theta) + \frac{K_{\text{ref}}(T)}{D_T}\cos(\psi_{\text{ref}} - \theta)\right) \quad (6)$$

where $Z$ is the normalization constant, and $T$ denotes the final timestep. The proof is given in the Appendix. In the long-time limit, as the average phase $\psi(T)$ synchronizes toward the reference phase $\psi_{\text{ref}}$, the distribution further simplifies to a well-known von Mises form (Gaussian on a circle):

$$\texttt{von Mises Distribution:} \quad p_{\text{st}}(\theta) \approx \frac{1}{Z} \exp\left(\frac{K(T)r(T) + K_{\text{ref}}(T)}{D_T}\cos(\psi_{\text{ref}} - \theta)\right) \quad (7)$$

**Local Coupling.** A natural variant of the Kuramoto model is to introduce local coupling, where each oscillator interacts only with its neighboring oscillators:

$$\frac{d\boldsymbol{\theta}_t^i}{dt} = \frac{1}{|\mathcal{N}_i|} \sum_{j \in \mathcal{N}_i} K(t)\sin(\boldsymbol{\theta}_t^j - \boldsymbol{\theta}_t^i) + K_{\text{ref}}(t)\sin(\psi_{\text{ref}} - \boldsymbol{\theta}_t^i) + \sqrt{2D_t}\xi^i \quad (8)$$

where $\mathcal{N}_i$ denotes the set of neighbors for the $i$-th oscillator. In contrast to the globally coupled case, local interactions introduce spatial inhomogeneity and break the mean-field approximation. As a result, the system exhibits diffusion-like behavior, with oscillatory phases progressively smoothing out over space and time. This manifests as blurring effects during the forward process, analogous to phenomena observed in heat dissipation models [51] and blurring diffusion models [24].

Interestingly, while local coupling prevents analytical simplification of the dynamics, the strong reference phase (via $K_{\text{ref}}$) still guides the system toward global synchronization. After sufficiently long evolution, the terminal distribution concentrates around the reference phase and can be effectively approximated by a von Mises distribution, despite the absence of mean-field symmetry.

**Coupling Strength and Noise Variance.** In the forward process, we maintain the relationship $K_{\text{ref}}(t) > D_t > K(t)$ to balance structure and noise. $D_t > K(t)$ ensures that the stochastic noise is strong enough to disrupt image details, while $K_{\text{ref}}(t) > D_t$ guarantees that the stronger reference coupling dominates over noise and local/global coupling to guide the system toward synchronization.

---

**Algorithm 1** Training algorithm for Kuramoto orientation diffusion models.

---

**Require:** Score prediction network $s(\cdot, \cdot)$, noise variance schedule $\{2D_t\}_{t=0}^{T-1}$, forward Kuramoto drift $f(\cdot, \cdot)$, noise $\epsilon \sim \mathcal{N}(0, I)$, number of MC samples $M$.

1: **repeat**
2:      Sample initial phase variable: $\boldsymbol{\theta}_0 \sim p(\boldsymbol{\theta}_0)$
3:      Sample a timestep: $t \sim U(1, T)$
4:      Simulate the forward Markov chain to obtain $\boldsymbol{\theta}_{t-1} \sim p(\boldsymbol{\theta}_{t-1}|\boldsymbol{\theta}_0)$
5:      Initialize sample counter $m = 0$
6:      **while** $m < M$ **do**
7:          Sample: $\boldsymbol{\theta}_t^m = \boldsymbol{\theta}_{t-1} + f(\boldsymbol{\theta}_{t-1}, t-1) + \sqrt{2D_{t-1}}\epsilon$
8:          Wrap phase: $\boldsymbol{\theta}_t^m = (\boldsymbol{\theta}_t^m + \pi) \bmod (2\pi) - \pi$
9:          Compute local score: $\nabla_{\boldsymbol{\theta}_t^m} \log p(\boldsymbol{\theta}_t^m|\boldsymbol{\theta}_{t-1})$
10:        $m \leftarrow m + 1$
11:      **end while**
12:      Compute training loss: $\mathcal{L} = \frac{1}{M} \sum_{m=0}^{M-1} \left( 2D_t \left\| s(\boldsymbol{\theta}_t^m, t) - \nabla_{\boldsymbol{\theta}_t^m} p(\boldsymbol{\theta}_t^m|\boldsymbol{\theta}_{t-1}) \right\|^2 \right)$
13: **until** converged

---

## 2.3 Learning the Score Function

In the reverse process (right-to-left in Fig. 2), we perform desynchronization guided by learned score functions, progressively restoring image complexity from a synchronized von Mises distribution back to diverse angular states. Below we illustrate how the score function is learned.

**Local Score Matching.** In our stochastic Kuramoto models, the presence of the nonlinear drift renders the marginal distribution $p(\boldsymbol{\theta}_t)$ intractable, and consequently the score function $\nabla_{\boldsymbol{\theta}_t} \log p(\boldsymbol{\theta}_t)$ is also unavailable in closed form. As a result, typical score-matching algorithms that rely on direct access to the marginal density cannot be applied. Nevertheless, we can still train a score network by exploiting the local Markov transition kernel $p(\boldsymbol{\theta}_t|\boldsymbol{\theta}_{t-1})$, based on the following general identity:

$$\nabla_{\boldsymbol{\theta}_t} \log p(\boldsymbol{\theta}_t) = \mathbb{E}_{\boldsymbol{\theta}_{t-1} \sim p(\boldsymbol{\theta}_{t-1}|\boldsymbol{\theta}_t)} \left[ \nabla_{\boldsymbol{\theta}_t} \log p(\boldsymbol{\theta}_t|\boldsymbol{\theta}_{t-1}) \right] \tag{9}$$

where the expectation is taken over the reverse transition $p(\boldsymbol{\theta}_{t-1}|\boldsymbol{\theta}_t)$. A detailed derivation of this identity is provided in the Appendix. Although the reverse transition is intractable in practice, Denoising Score Matching (DSM) [65] shows that the score function can be learned by sampling from the forward transition $p(\boldsymbol{\theta}_t|\boldsymbol{\theta}_{t-1})$ instead. Specifically, the training objective can be written as:

$$\mathbb{E}_{t \sim U(1,T)} \mathbb{E}_{\boldsymbol{\theta}_0} \mathbb{E}_{\boldsymbol{\theta}_{t-1} \sim p(\boldsymbol{\theta}_{t-1}|\boldsymbol{\theta}_0)} \mathbb{E}_{\boldsymbol{\theta}_t \sim p(\boldsymbol{\theta}_t|\boldsymbol{\theta}_{t-1})} \left[ 2D_t \| s(\boldsymbol{\theta}_t, t) - \nabla \log p(\boldsymbol{\theta}_t|\boldsymbol{\theta}_{t-1}) \|^2 \right] \tag{10}$$

where $s(\boldsymbol{\theta}_t, t)$ denotes the score network being optimized. At each training step, we simulate the forward Markov chain to obtain $\boldsymbol{\theta}_{t-1}$, and then sample multiple $\boldsymbol{\theta}_t$ from the local transition kernel $p(\boldsymbol{\theta}_t|\boldsymbol{\theta}_{t-1})$ to estimate the training loss via Monte Carlo (MC) approximations.

**Wrapped Gaussian Transition.** Due to the phase wrapping on a periodic domain, the local transition probability $p(\boldsymbol{\theta}_t|\boldsymbol{\theta}_{t-1})$ follows a wrapped Gaussian distribution:

$$p(\boldsymbol{\theta}_t|\boldsymbol{\theta}_{t-1}) = \mathcal{WN}\left( \boldsymbol{\theta}_{t-1} + \boldsymbol{f}(\boldsymbol{\theta}_{t-1}, t-1), 2D_{t-1}\boldsymbol{I} \right)$$

$$\approx \frac{1}{\sqrt{4\pi D_{t-1}}} \sum_{k=-K}^{K} \exp \frac{-\left( \boldsymbol{\theta}_t - \boldsymbol{\theta}_{t-1} - \boldsymbol{f}(\boldsymbol{\theta}_{t-1}, t-1) + 2\pi k \right)^2}{4D_{t-1}} \tag{11}$$

where $\boldsymbol{f}(\boldsymbol{\theta}_t, t)$ denotes the forward drift of the Kuramoto model, *i.e.,* the coupling and reference terms weighted by the coupling strength. Since the wrapped Gaussian involves an infinite series, neither the transition density nor its score function admits a simple closed-form. We thus approximate the transition by truncating the summation to a small finite number of terms $K$.

**Periodicity-aware Networks.** To incorporate the inherent periodicity of the phase variables into the score network, we embed the input phases using sinusoidal features $[\sin(\boldsymbol{\theta}), \cos(\boldsymbol{\theta})]$ as input to the

Table 1: FID Scores (↓) on SOCOFing fingerprint dataset [56].

| Diffusion Steps | 100 | 300 | 1000 |
|---|---|---|---|
| SGM [60] | 104.92 | 62.66 | 23.84 |
| Kuramoto Orientation Diffusion (Globally Coupled) | 74.41 | 47.93 | 20.64 |
| Kuramoto Orientation Diffusion (Locally Coupled) | 67.49 | 43.57 | 18.75 |

Table 2: FID Scores (↓) on Brodatz texture dataset [1, 8].

| Steps | 100 | 300 | 1000 |
|---|---|---|---|
| SGM [60] | 38.33 | 22.40 | 20.37 |
| Kuramoto Orientation Diffusion (Globally Coupled) | 20.26 | 18.51 | 15.42 |
| Kuramoto Orientation Diffusion (Locally Coupled) | 18.47 | 15.93 | 14.19 |

network. The network predicts two outputs $[s_1(\boldsymbol{\theta}, t), s_2(\boldsymbol{\theta}, t)]$ corresponding to the two Cartesian components. We then project the output back onto the angular domain via:

$$s(\boldsymbol{\theta}, t) = s_1(\boldsymbol{\theta}, t) \cos(\boldsymbol{\theta}) + s_2(\boldsymbol{\theta}, t) \sin(\boldsymbol{\theta}) \tag{12}$$

This operation ensures that the score prediction respects the circular geometry of the phase space.

**Training Algorithms.** We summarize the training algorithms in Alg. 1. To adapt non-periodic image data to the periodic phase domain, input pixels in the range $[-1, 1]$ are linearly mapped to $[-0.9\pi, 0.9\pi]$. This margin near the boundaries helps distinguish values near $-1$ and $1$, which would otherwise collapse under phase wrapping. Throughout both training and inference, we enforce the periodic geometry by wrapping all phase variables into the interval $[-\pi, \pi]$ after each SDE step. Our training procedure relies on estimating local scores using Monte Carlo samples from truncated wrapped Gaussian transitions. We find that using $K=3$ and $M=5$ samples per step provides a good balance between training stability and computational efficiency.

## 3 Experiments

### 3.1 Setup

**Datasets, Baselines, and Metrics.** We evaluate our proposed method across both general-purpose and orientation-rich image generation tasks. For standard benchmarking, we first test on CIFAR10 [35]. To specifically assess performance on orientation-dense data, we then apply our model to the SOCOFing fingerprint dataset [56], the Brodatz texture dataset [1, 8], and the ground terrain dataset [68]. The input image resolutions are $3\times32\times32$ for CIFAR10 and $1\times96\times96$ for SOCOFing. The Brodatz

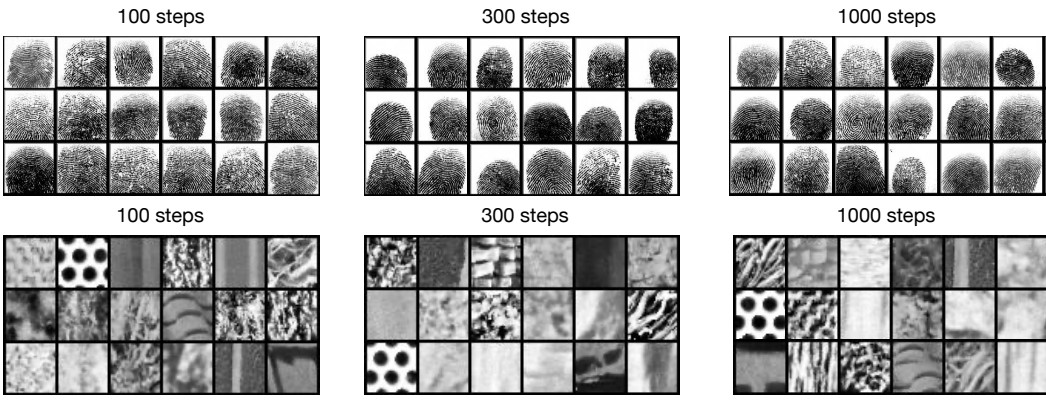

Figure 3: Samples generated by our Kuramoto orientation diffusion model on SOCOFing fingerprint and Brodatz texture datasets under varying denoising steps.

Table 3: FID Scores (↓) on the ground terrain dataset [68].

| Diffusion Steps | 100 | 300 | 1000 |
|---|---|---|---|
| SGM [60] | 114.90 | 56.72 | 33.79 |
| Kuramoto Orientation Diffusion (Globally Coupled) | 101.65 | 54.17 | 33.56 |
| Kuramoto Orientation Diffusion (Locally Coupled) | 92.86 | 49.68 | 30.62 |

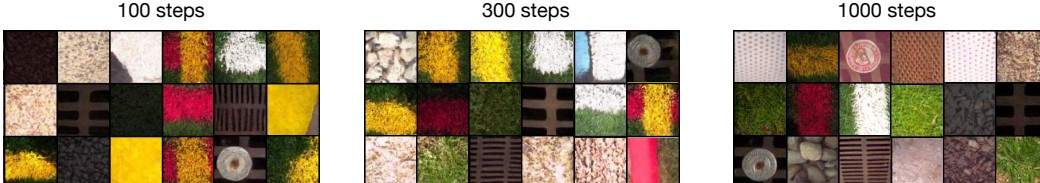

Figure 4: Samples generated by our Kuramoto model on the ground terrain dataset.

texture dataset originally consists of high-resolution samples depicting various textures (*e.g.,* grass, water, sand, wool) with sizes of $1 \times 512 \times 512$ or $1 \times 1024 \times 1024$. Due to the limited number of available images, we increase the dataset size by dividing these high-resolution textures into smaller patches of $1 \times 32 \times 32$. Compared to Brodatz textures, the ground terrain dataset [68] contains a broader set of orientation-rich material textures (*e.g.,* plastic, turf, steel, asphalt, leaves, stone, and brick) at the higher resolution of $128 \times 128$. We primarily compare with the standard variance-preserving (VP) score-based generative model (SGM) [60]. To evaluate the quality of generated images, we use the Fréchet Inception Distance (FID) [21], a widely used metric for assessing visual fidelity and diversity. All models are trained until the FID score converges to ensure fair comparison.

In the Supplementary, we extend evaluation beyond images to (i) Earth and climate science datasets on the 2D Sphere [46, 45, 5, 18], and (ii) Navier-Stokes fluid velocity fields [6]. The former suite of datasets is defined on *naturally periodic grids*, while the latter one constitutes *intrinsically angular data*. Together with the image datasets, the benefit of the synchronization inductive bias built into our model will be consistently validated across three distinct domains – standard orientation-dense images (with pixels mapped to phases), spherical geophysical fields, and fluid velocity phases.

**Timesteps.** Due to the phase coupling in our Kuramoto diffusion model, the forward process leads to faster convergence toward the terminal distribution compared to conventional diffusion models. To explore this advantage, we evaluate our model under varying diffusion step counts: 100, 300, and 1000 steps. These comparisons reveal how efficiently our model captures structural information with fewer steps. Details of the noise and coupling schedules are provided in the Supplementary.

## 3.2 Results

**Fingerprints and Textures.** Tables 1 and 2 report FID scores on the SOCOFing fingerprint and Brodatz texture datasets, respectively. Across all diffusion step settings, both globally and locally coupled variants of our Kuramoto orientation diffusion model consistently outperform the standard SGM [60]. The improvement is especially notable on the Brodatz texture dataset, where our 100-step Kuramoto model achieves performance comparable to or better than SGM using 1000 steps – demonstrating a substantial gain in sampling efficiency. These results underscore the advantage of incorporating structured coupling dynamics for modeling orientation-dense data. The synchronization-based inductive bias helps preserve coherent directional patterns, which are critical for the perceptual quality of textures and fingerprints. Notably, the locally coupled variant offers further improvements by aligning with the local spatial correlations. Fig. 3 provides qualitative examples, illustrating that our method produces sharp and consistent patterns under varying diffusion steps.

**Ground Terrain.** Table 3 reports FID on the ground terrain dataset [68], comparing our Kuramoto diffusion model with SGM. The trend mirrors fingerprints and textures: across denoising step counts, our method consistently attains lower FID. This indicates that the synchronization prior continues to help at orientation-dense scenes of higher resolutions. Qualitative results in Fig. 4 illustrate that,

Table 4: FID Scores (↓) on CIFAR10 [35].

| Diffusion Steps | 100 | 300 | 1000 |
|---|---|---|---|
| SGM [60] | 38.04 | 25.76 | 3.17 |
| Kuramoto Orientation Diffusion (Globally Coupled) | 29.96 | 25.83 | 11.58 |
| Kuramoto Orientation Diffusion (Locally Coupled) | 28.17 | 24.86 | 10.79 |

100 steps        300 steps        1000 steps

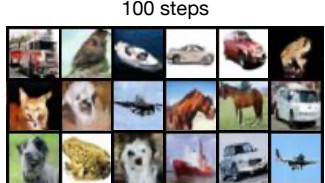 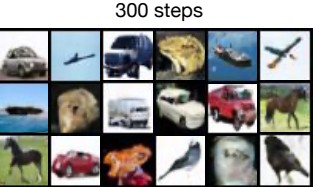 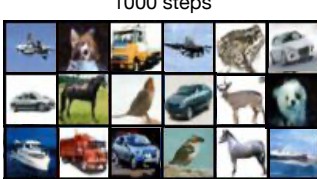

Figure 5: Samples generated by our Kuramoto model on CIFAR10 under varying denoising steps.

even under fewer denoising steps, our samples exhibit coherent directional structure and material appearance, while additional steps further refine detail realism and texture alignment.

**CIFAR10.** Table 4 presents the FID scores for our Kuramoto orientation diffusion models against SGM on CIFAR10. At 100 diffusion steps, both Kuramoto models substantially outperform SGM, highlighting the effectiveness of structured synchronization as an inductive bias in low-step regimes. At 300 steps, Kuramoto models achieve comparable or slightly better performance than SGM, demonstrating their ability to maintain sample quality as diffusion progresses. At 1000 steps, SGM achieves the best overall score, though both Kuramoto models remain competitive.

Fig. 5 displays generated samples under different step counts. As diffusion steps increase, the benefits of structured synchronization become less prominent. We expect that in longer trajectories, the structured reverse dynamics may inhibit the flexibility to fully capture fine-grained details, especially in datasets that lack strong orientation priors. These results suggest that Kuramoto-based synchronization dynamics are particularly advantageous in generating high-quality images within limited steps, even on general-purpose datasets. However, this structured bias may slightly limit expressiveness under excessive denoising steps on natural images lacking strong orientation patterns.

## 3.3 Discussions

**Structured Destruction.** Our Kuramoto forward process introduces a very structured destruction process, leveraging either global or local coupling mechanisms to achieve more controlled noising dynamics. Unlike conventional isotropic diffusion which quickly loses object structure, our model preserves the objects in the early stages through synchronized coupling (see also Fig. 1). This structured noising progressively aligns similar phase variables while maintaining orientation consistency, allowing the model to better retain structural information. As noise levels increase, the coupling interactions expedite the convergence to the noise distribution, enabling a faster transition compared to standard diffusion models. This unique dual-phase dynamic – **initial structured synchronization followed by rapid noise adaptation** – allows our model to converge faster while maintaining orientation coherence. The empirical SNR plot in Fig. 6 clearly demonstrates the advantage described above, highlighting how the coupling-driven dynamics allow for efficient and structured noise progression.

**Hierarchical Generation Process.** Figs. 7 and 8 depict the hierarchical generation process of our Kuramoto orientation diffusion model. The generation starts from a synchronized state sampled from the von Mises distribution, representing a low-entropy configuration with highly aligned phase variables. In the reverse process, the large-scale structure is established first, as global coherence from the initial synchronized state is preserved. This occurs because the forward diffusion inherently maintains global coherence in the later stages, which correspond to the early stages in the reverse process. Once the primary structure is set, finer-scale details gradually emerge through successive diffusion steps, driven by anti-coupling dynamics that introduce local variability, allowing the image to evolve into more complex and nuanced patterns. Our model follows a coarse-to-fine paradigm, where the global structure remains consistent while localized, intricate features evolve flexibly.

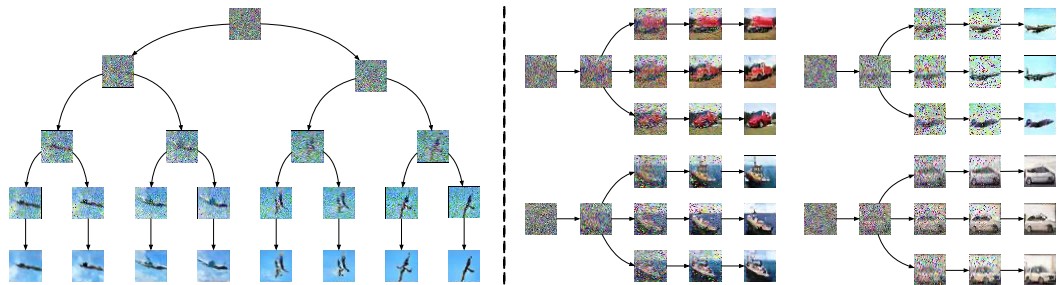

Figure 7: Hierarchical generation process of our locally coupled Kuramoto diffusion model applied to CIFAR-10. The generation follows a coarse-to-fine progression, starting from a structured von Mises sample and progressively adding fine-scale details. These results correspond to 100 diffusion steps.

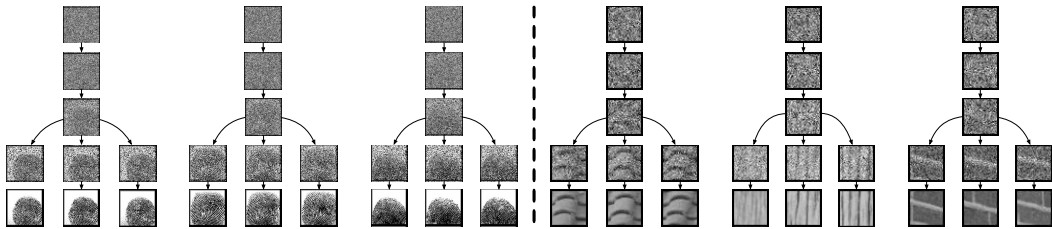

Figure 8: Hierarchical generation of our locally coupled Kuramoto diffusion model on the SOCOFing fingerprint dataset (left) and Brodatz texture dataset (right). The model first establishes large-scale orientation patterns, followed by finer texture details. These results correspond to 300 diffusion steps.

The hierarchical nature of the Kuramoto diffusion enhances interpretability by providing a clear generative pathway from **global coherence** to **local variability**. This hierarchical generation approach aligns with the spectral biases observed in coarse-to-fine generation of diffusion models [34, 51]. In typical score-based diffusion models, this hierarchy arises implicitly from the progressive noise attenuation, preserving low-frequency components longer while high-frequency details emerge later. In contrast, Rissanen *et al.* [51] use heat equations for the forward process, resulting in isotropic blurring as high-frequency components dissipate. Our Kuramoto model explicitly encodes hierarchical progression through synchronization dynamics. This structured, non-isotropic phase alignment offers a more interpretable pathway from global to local patterns.

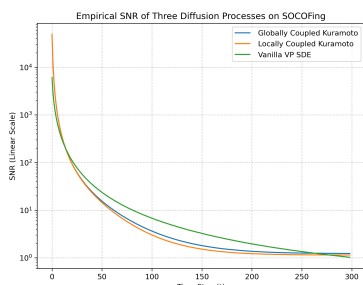

Figure 6: Empirical SNR over time.

## 4 Related Work

**Generative Models.** Deep generative models have made remarkable progress in the last decade, both in the creation of powerful foundation models [52] and in the design of sample-efficient optimization algorithms [66, 49]. Early advances in deep generative modeling, such as Variational Autoencoders (VAEs) [33] and Generative Adversarial Networks (GANs) [19], pushed the frontiers of generative capabilities but faced distinct limitations: VAEs often produced blurry samples, while GANs suffered from instability in training. Recently, diffusion models [57, 23, 60, 58, 13, 25, 32] have emerged as a powerful, principled alternative. These models learn to reverse a noising process by estimating the score function $\nabla_{\boldsymbol{x}} \log p(\boldsymbol{x}_t)$ of intermediate Gaussian-corrupted distributions. Building on the view of generative modeling as a continuous transformation from noise to data, Flow Matching [38] and Rectified Flow [39] directly learn velocity fields connecting noise and data without relying on stochastic diffusion, sidestepping the need for score estimation. Stochastic Interpolants [2] further generalize diffusion and flow models by learning stochastic trajectories interpolating between the data and prior, providing a unifying perspective across two modeling paradigms. Several recent

efforts extend flow matching and diffusion models to non-Euclidean geometries. For example, Riemannian Flow Matching [10] formulates generative models directly on Riemannian manifolds, respecting underlying geometric constraints. Other Riemannian diffusion models account for manifold curvature to enable generative modeling over geometries such as hyperspheres, tori, and hyperbolic spaces [11, 37, 26, 12, 69, 28, 14]. These advances highlight the growing interest in incorporating geometric inductive biases into generative modeling.

Our work contributes to this landscape by exploring nonlinear diffusion models grounded in stochastic Kuramoto dynamics. Instead of linear drifts in conventional diffusion models, we introduce structured phase dynamics driven by nonlinear Kuramoto coupling, evolving on the periodic domain. Unlike classical score-based models that assume a Gaussian forward process, our framework leads to non-Gaussian, wrapped distributions that evolve according to nonlinear SDEs. This extends recent interest in geometry-aware generative models and interpretable generation process.

**Neural Oscillations in Machine Learning.** Oscillatory dynamics are central to understanding biological neural systems, where phase-locked rhythms and traveling waves are believed to support functions such as sensory binding, working memory, and attention [9, 42, 16]. Motivated by their computational relevance, recent machine learning research has increasingly incorporated oscillatory and wave-based dynamics into neural representations, treating them as inductive biases that promote generalization and structured behaviors [54, 55, 15, 61, 30, 31]. A key focus has been on **synchronization**, where multiple units align their oscillatory phases to form stable or emergent patterns. This phenomenon has been studied through the lens of the Kuramoto model [36], a classical framework from nonlinear dynamics that describes how populations of coupled oscillators evolve toward synchronization. In computational neuroscience and machine learning, Kuramoto-inspired models have been applied to tasks such as modeling neural connectivity [7], clustering through emergent synchrony [50], and mitigating over-smoothing in graph neural networks [44]. A recent notable example is the Artificial Kuramoto Oscillatory Neuron (AKOrN) framework [41], which replaces thresholding units with oscillatory ones to bind neurons together through synchronization dynamics. These efforts reflect the growing interest in integrating principles of neural oscillations and synchronization into machine learning frameworks, offering novel perspectives and tools for modeling complex, dynamic systems.

# 5 Conclusion

This paper introduces a nonlinear score-based generative framework that incorporates stochastic Kuramoto dynamics to model orientation-rich data. By formulating the forward process as synchronization and the reverse process as desynchronization, our method brings biologically inspired inductive biases into the diffusion process. Through wrapped Gaussian transitions and periodicity-aware networks, the model naturally captures the geometry of angular data. Experiments show that our approach outperforms conventional baselines on orientation-dense datasets and remains competitive on general image generation tasks. This work highlights the potential of integrating biologically inspired synchronization dynamics as structured priors in generative modeling of orientation-dense data, paving the way for incorporating nonlinear dynamics into generative models.

**Limitations and Future Work.** A primary limitation lies in the training efficiency: each training step incurs an $\mathcal{O}(T)$ time cost due to the need to explicitly simulate the forward Markov chain. Actually, this cost actually can be nearly eliminated with ***pre-computation & cache***: before training, we can run the forward SDE on the entire dataset, save all pairs to disk, and then load them directly during training. This makes the simulation cost effectively $\mathcal{O}(0)$ at each training step. Otherwise if disk space is limited, we can simply re-generate and cache one epoch's worth of pairs at the start of each epoch. This still dramatically reduces the simulation overhead.

Beyond computational improvements, applying this framework to neural spiking data offers a compelling opportunity to further explore the biological plausibility of synchronization-driven generative models and to bridge the gap between theoretical neuroscience and machine learning.

**Broader Impacts.** Our approach may benefit socially relevant domains such as biometric security (*e.g.,* synthetic fingerprint generation), medical imaging (*e.g.,* fiber orientation modeling in MRI), and scientific visualization (*e.g.,* materials analysis). As with other generative technologies, we advocate for responsible research and deployment in accordance with ethical and societal guidelines.

## Acknowledgments

We would like to thank anonymous reviewers for their constructive suggestions and feedback. Yue Song was supported by the gift donation from Cisco. T. Anderson Keller was supported by the Kempner Institute Research Fellowship. Takeru Miyato was supported by the ERC Starting Grant LEGO-3D (850533) and the German Research Foundation (DFG): SFB 1233, Robust Vision: Inference Principles and Neural Mechanisms, project number: 276693517.

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

# A   Math Derivations and Intuitions

## A.1   Mean-field Dynamics of Fokker-Planck Equation for Stochastic Kuramoto Models

In the thermodynamic limit of infinite oscillators ($N \to \infty$), the collective dynamics of the oscillator population can be effectively described by the mean-field evolution of a single representative oscillator with phase variable $\theta$. The corresponding probability density $p(\theta, t)$ evolves according to the Fokker–Planck equation:

$$\frac{\partial p(\theta, t)}{\partial t} = -\frac{\partial}{\partial \theta}\Big[\big(K(t)r(t)\sin(\psi(t) - \theta) + K_{\text{ref}}(t)\sin(\psi_{\text{ref}} - \theta)\big)p(\theta, t)\Big] + D_t \frac{\partial^2 p(\theta, t)}{\partial \theta^2} \tag{13}$$

where $r(t)$ and $\psi(t)$ denote the magnitude and phase of the complex order parameter, respectively. In the long-time limit, the quasi-equilibrium satisfies $\partial p_{\text{st}}(\theta)/\partial t \approx 0$. At this point, the stationary solution $p(\theta, T)$ approximately follows:

$$\begin{aligned}
D_T \frac{\partial^2 p(\theta, T)}{\partial \theta^2} &\approx \frac{\partial}{\partial \theta}\Big[\big(K(T)r(T)\sin(\psi(T) - \theta) + K_{\text{ref}}(T)\sin(\psi_{\text{ref}} - \theta)\big)p(\theta, T)\Big] \\
D_T \frac{\partial p(\theta, T)}{\partial \theta} &\approx \big(K(T)r(T)\sin(\psi - \theta) + K_{\text{ref}}(T)\sin(\psi_{\text{ref}} - \theta)\big)p(\theta, T) + C_1 \\
\frac{1}{p(\theta, T)}\frac{\partial p(\theta, T)}{\partial \theta} &\approx \frac{K(T)r(T)\sin(\psi(T) - \theta) + K_{\text{ref}}(T)\sin(\psi_{\text{ref}} - \theta)}{D_T} \\
\log p(\theta, T) &\approx \frac{K(T)r(T)}{D_T}\cos(\psi(T) - \theta) + \frac{K_{\text{ref}}(T)}{D_T}\cos(\psi_{\text{ref}} - \theta) + C_2 \\
p(\theta, T) &\approx \frac{1}{Z}\exp\big(\frac{K(T)r(T)}{D_T}\cos(\psi(T) - \theta) + \frac{K_{\text{ref}}(T)}{D_T}\cos(\psi_{\text{ref}} - \theta)\big)
\end{aligned} \tag{14}$$

where $C_1, C_2$ are constants, and $Z$ is the normalization constant. We assume $C_1 = 0$ because the distribution is periodic, and $C_2$ is absorbed into $Z$. Since the average phase $\psi(T)$ will synchronize to the reference $\psi_{\text{ref}}$, the quasi-stationary solution is given by:

$$p_{\text{st}}(\theta) \approx \frac{1}{Z}\exp\left(\frac{K(T)r(T) + K_{\text{ref}}(T)}{D_T}\cos(\psi_{\text{ref}} - \theta)\right) \tag{15}$$

This final form follows a von Mises distribution, reflecting the low-entropy steady state induced by synchronized interactions and reference phase attraction.

## A.2 Local Score Matching

At any timestep $t$ of the Markov chain, the marginal distribution $p(\boldsymbol{\theta}_t)$ can be expressed as:

$$p(\boldsymbol{\theta}_t) = \int p(\boldsymbol{\theta}_{t-1}) p(\boldsymbol{\theta}_t|\boldsymbol{\theta}_{t-1}) \, d\boldsymbol{\theta}_{t-1}$$

$$\nabla_{\boldsymbol{\theta}_t} p(\boldsymbol{\theta}_t) = \int p(\boldsymbol{\theta}_{t-1}) \nabla_{\boldsymbol{\theta}_t} p(\boldsymbol{\theta}_t|\boldsymbol{\theta}_{t-1}) \, d\boldsymbol{\theta}_{t-1} \tag{16}$$

For any score $\nabla_{\boldsymbol{\theta}_t} \log p(\boldsymbol{\theta}_t)$, we have $\nabla_{\boldsymbol{\theta}_t} \log p(\boldsymbol{\theta}_t) = \nabla_{\boldsymbol{\theta}_t} p(\boldsymbol{\theta}_t) / p(\boldsymbol{\theta}_t)$. Then we can rewrite the above equation leveraging this identity:

$$p(\boldsymbol{\theta}_t) \nabla_{\boldsymbol{\theta}_t} \log p(\boldsymbol{\theta}_t) = \int p(\boldsymbol{\theta}_{t-1}) p(\boldsymbol{\theta}_t|\boldsymbol{\theta}_{t-1}) \nabla_{\boldsymbol{\theta}_t} \log p(\boldsymbol{\theta}_t|\boldsymbol{\theta}_{t-1}) \, d\boldsymbol{\theta}_{t-1}$$

$$\cancel{p(\boldsymbol{\theta}_t)} \nabla_{\boldsymbol{\theta}_t} \log p(\boldsymbol{\theta}_t) = \int \cancel{p(\boldsymbol{\theta}_t)} p(\boldsymbol{\theta}_{t-1}|\boldsymbol{\theta}_t) \nabla_{\boldsymbol{\theta}_t} \log p(\boldsymbol{\theta}_t|\boldsymbol{\theta}_{t-1}) \, d\boldsymbol{\theta}_{t-1} \tag{17}$$

$$\nabla_{\boldsymbol{\theta}_t} \log p(\boldsymbol{\theta}_t) = \mathbb{E}_{\boldsymbol{\theta}_{t-1} \sim p(\boldsymbol{\theta}_{t-1}|\boldsymbol{\theta}_t)} \left[ \nabla_{\boldsymbol{\theta}_t} \log p(\boldsymbol{\theta}_t|\boldsymbol{\theta}_{t-1}) \right]$$

The above results indicate that the score can be effectively approximated by the local transition kernel.

## A.3 Fourier Interpretation of Kuramoto Coupling

From a spectral standpoint, local phase coupling behaves like an angular low-pass filter. In the small-angle approximation ($\sin(\boldsymbol{\theta}_j - \boldsymbol{\theta}_i) \approx \boldsymbol{\theta}_j - \boldsymbol{\theta}_i$) and without a global reference phase, the forward SDE reduces to a stochastic heat equation (or a graph Laplacian form equivalently):

$$\frac{d\boldsymbol{\theta}_t^i}{dt} = K(t) \nabla^2 \boldsymbol{\theta}_t^i + \sqrt{2D_t} \xi^i \tag{18}$$

where $\nabla^2$ denotes the Laplacian operator. Each Fourier component decays like $e^{-K(t)k^2 t}$, where $k$ denotes the spatial frequency. Modes with very high spatial frequency $k$ (pixel-scale noise) are damped almost instantly, while moderate-frequency modes (coherent, edge-defining structures) decay much more slowly. Consequently, the Kuramoto drift effectively filters away pixel-scale noise while preserving the smooth, orientation-rich patterns that constitute edges.

## A.4 Why Kuramoto Diffusion Falls Short on General Natural Images

Our Kuramoto coupling is foremost a synchronization mechanism: it excels at pulling similar phases together, thereby preserving edges and repetitive patterns. On orientation-rich data where the data are dominated by the **simple outlines and repeating ridge structures**, this yields sharp samples in fewer steps. By contrast, natural images (*e.g.,* CIFAR-10) demand modeling ***e.g.,* complex, global semantics (object shapes, color gradients, backgrounds), often characterized by higher-order and longer-range correlations**. In this setting, the local synchronization bias becomes less relevant and potentially even detrimental; an isotropic diffusion process with a global drift may be better suited to capture these large-scale, non-repetitive features. As a result, our method trades some global fidelity (reflected in higher CIFAR-10 FID) in exchange for strong local coherence.

# B    More Experimental Results

## B.1 Implementation Details

Alg. 2 outlines the inference procedure for our model. We adopt the SDE formulation by default, but it can be optionally replaced with an ODE solver for improved efficiency by modifying Line 4 to: $\boldsymbol{\theta}_{t-1} = \boldsymbol{\theta}_t - \boldsymbol{f}(\boldsymbol{\theta}_t, t) + D_t \cdot s(\boldsymbol{\theta}_t, t)$. We use AdamW optimizer with a learning rate of $1e{-}4$ and apply exponential moving average (EMA) updates with decay rate $0.995$. A single NVIDIA A100 GPU is used for all the training and inference processes.

**Network Architectures.** We use a U-Net architecture following the design of [13, 60], equipped with three self-attention layers [63]. These are applied at spatial resolutions of $16$, $8$, and $4$ for CIFAR-10

**Algorithm 2** Inference algorithm for Kuramoto orientation diffusion models.

---

**Require:** Trained score network $s(\cdot,\cdot)$, noise variance schedule $\{2D_t\}_{t=1}^{T}$, forward Kuramoto drift
$\quad \boldsymbol{f}(\cdot,\cdot)$, noise $\epsilon \sim \mathcal{N}(0,I)$.
1: Sample initial phase: $\boldsymbol{\theta}_T \sim p(\boldsymbol{\theta}_T)$
2: Initialize timestep counter $t = T$
3: **while** $t > 0$ **do**
4: $\quad$ Update: $\boldsymbol{\theta}_{t-1} = \boldsymbol{\theta}_t - \boldsymbol{f}(\boldsymbol{\theta}_t, t) + 2D_t \cdot s(\boldsymbol{\theta}_t, t) + \sqrt{2D_t}\epsilon$
5: $\quad$ Wrap phase: $\boldsymbol{\theta}_{t-1} = (\boldsymbol{\theta}_{t-1} + \pi) \bmod (2\pi) - \pi$
6: $\quad$ $t \leftarrow t - 1$
7: **end while**

---

and Brodatz, and 24, 12, and 6 for SOCOFing. Timestep conditioning is implemented via sinusoidal positional embeddings. Each block uses group normalization [67] and GELU activations [20] throughout. The same architecture is shared across both our method and SGM for fair comparison.

Table 5: Linear schedules of noise variance and coupling strength under varying diffusion steps.

| Steps | Global Coupling | | | Local Coupling | | |
|---|---|---|---|---|---|---|
| | $2D$ | $K$ | $K_{\text{ref}}$ | $2D$ | $K$ | $K_{\text{ref}}$ |
| 100 | $[1e{-}4, 0.1]$ | $[3e{-}5, 0.03]$ | $[4.5e{-}5, 0.045]$ | $[1e{-}4, 0.1]$ | $[5e{-}5, 0.05]$ | $[5e{-}5, 0.05]$ |
| 300 | $[1e{-}4, 0.07]$ | $[3e{-}5, 0.02]$ | $[4.5e{-}5, 0.03]$ | $[1e{-}4, 0.07]$ | $[5e{-}5, 0.03]$ | $[5e{-}5, 0.03]$ |
| 1000 | $[1e{-}4, 0.015]$ | $[3e{-}5, 0.0045]$ | $[4.5e{-}5, 0.00675]$ | $[1e{-}4, 0.025]$ | $[5e{-}5, 0.01]$ | $[5e{-}5, 0.01]$ |

**Schedules of Noise and Coupling Strength.** Table 5 summarizes the linear schedules for the noise variance $2D_t$, internal coupling strength $K$, and reference phase coupling $K_{\text{ref}}$. In the globally coupled variant, we maintain the relation $K_{\text{ref}} > 2D > K$ to ensure synchronization toward a reference phase while allowing sufficient noise injection. In the locally coupled variant, we increase the internal coupling strength $K$ to compensate for its restricted spatial influence, which is limited to a $5 \times 5$ neighborhood around each pixel. We keep $K_{\text{ref}} = K$ in this case, since the reference coupling term inherently has a broader impact by acting across the entire image domain. For the SGM baseline, we adopt the VP-SDE formulation with linear variance schedules: $[1e{-}4, 0.1]$ for 100 steps, $[1e{-}4, 0.07]$ for 300 steps, and $[1e{-}4, 0.02]$ for 1000 steps.

## B.2 Ablation Study and Alternative Metrics

Table 6: FID Scores ($\downarrow$) on Brodatz texture dataset [1, 8].

| Steps | 100 | 300 | 1000 |
|---|---|---|---|
| SGM [60] | 38.33 | 22.40 | 20.37 |
| Reference-only Process ($K(t) = 0$) | 33.76 | 20.54 | 19.01 |
| Kuramoto Orientation Diffusion (Globally Coupled) | 20.26 | 18.51 | 15.42 |
| Kuramoto Orientation Diffusion (Locally Coupled) | 18.47 | 15.93 | 14.19 |

**Ablation of the Kuramoto Coupling.** To isolate the impact of Kuramoto coupling, we include a model variant by setting $K(t) = 0$ in Eq. (3), which keeps only the phase-reference drift in the SDE:

$$\frac{d\boldsymbol{\theta}_t^i}{dt} = K_{\text{ref}}(t) \sin(\psi_{\text{ref}} - \boldsymbol{\theta}_t^i) + \sqrt{2D_t}\xi^i \tag{19}$$

Table 6 presents the evaluation results on Brodatz textures. Removing the non-linear Kuramoto coupling raises 100-step FID by over 15 points, which confirms that non-isotropic phase synchronization is the key driver of both rapid convergence and thhe performance gain on orientation-rich data.

**Alternative Metric.** The FID metric is known for several limitations – its dependence on Inception embeddings, its Gaussian-moment matching assumption, and occasional misalignment with human judgment. As a more robust alternative, we adopt CLIP-MMD (CMMD) [29], which computes a nonparametric Maximum Mean Discrepancy in the pretrained CLIP feature space, which combines the benefits of distribution-free sample efficiency with the strong embedding power of CLIP.

Table 7: CMMD Scores (↓) on Brodatz texture dataset [1, 8].

| Steps | 100 | 300 | 1000 |
|---|---|---|---|
| SGM [60] | 0.183 | 0.165 | 0.141 |
| Kuramoto Orientation Diffusion (Locally Coupled) | 0.072 | 0.045 | 0.030 |

Table 7 evaluates our local Kuramoto model versus SGM on the Brodatz textures dataset using the CMMD score. Our model achieves substantially lower CMMD at every step count. **Remarkably, the 100-step Kuramoto model outperforms the 1000-step SGM by a large margin.** These results mirror and amplify our FID improvements, demonstrating the superior sample fidelity even under a distribution-free, CLIP-based evaluation.

## B.3 Earth and Climate Science Datasets on Spheres

To further evaluate our Kuramoto orientation diffusion model on naturally periodic data, we consider four real-world datasets of Earth and climate events: significant volcanic eruptions, earthquakes, floods, and wildfires [46, 45, 5, 18]. These datasets capture empirical spatial distributions of geophysical events on the surface of the Earth and are inherently defined over a 2D spherical domain. We map longitude to $[-\pi, \pi]$ and linearly scale latitude from $\left[-\frac{\pi}{2}, \frac{\pi}{2}\right]$ to the same interval to ensure compatibility with the periodic modeling framework. We compare our approach against a suite of Riemannian geometry-aware generative models, including Riemannian CNFs [40], Moser flows [53], CNF matching [4], and Riemannian score-based or flow-based generative models [26, 12, 10].

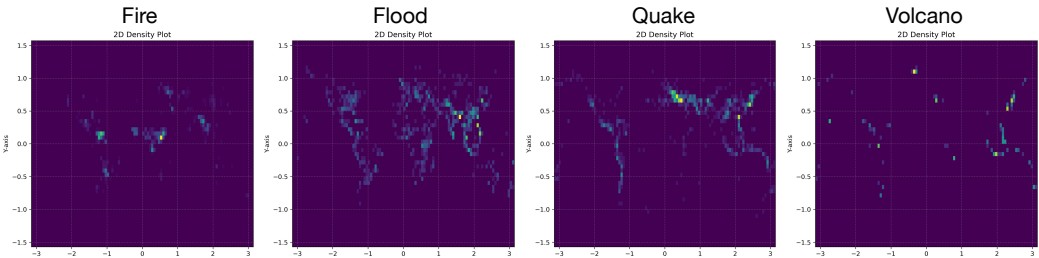

Figure 9: Learned density plot of our method on each Earth and climate science dataset. The X-axis denotes the longitude while the Y-axis represents the latitude.

Table 8: Test NLL on Earth and climate science datasets averaged across 5 runs.

| Dataset | Volcano | Earthquake | Flood | Fire |
|---|---|---|---|---|
| Riemannian CNF [40] | -6.05±0.61 | 0.14±0.23 | 1.11±0.19 | -0.80±0.54 |
| Moser Flow [53] | -4.21±0.17 | -0.16±0.06 | 0.57±0.10 | -1.28±0.05 |
| CNF Matching [4] | -2.38±0.17 | -0.38±0.01 | 0.25±0.02 | -1.40±0.02 |
| Riemannian score-based [12] | -4.92±0.25 | -0.19±0.07 | 0.48±0.17 | -1.33±0.06 |
| Riemannian diffusion model [26] | -6.61±0.96 | -0.40±0.05 | 0.43±0.07 | -1.38±0.05 |
| Riemannian flow matching [10] | -7.93±1.67 | -0.28±0.08 | 0.42±0.05 | -1.86±0.11 |
| Our Kuramoto orientation diffusion model | -5.18±0.17 | -0.18±0.06 | 0.49±0.18 | -1.44±0.05 |

Fig. 9 visualizes the learned densities produced by our method, capturing both highly concentrated regions (*e.g.,* volcanic and fire clusters) and dispersed patterns. Table 8 presents the test negative log-likelihood (NLL) on each dataset. Our method achieves comparable performance against these baselines. We compute NLL using the change-of-variables formula under the time-reversed ODE solver: $\log p(\boldsymbol{\theta}_0) = \log p(\boldsymbol{\theta}_T) + \sum_{t=T}^{1} \text{Tr}(\mathcal{J}_{\boldsymbol{b}(\boldsymbol{\theta}_t, t)})$ where $p(\boldsymbol{\theta}_T)$ denotes the von Mises prior and $\boldsymbol{b}(\boldsymbol{\theta}_t, t)$ denotes the backward drift, *i.e.,* $\boldsymbol{b}(\boldsymbol{\theta}_t, t) = -\boldsymbol{f}(\boldsymbol{\theta}_t, t) + D_t \cdot s(\boldsymbol{\theta}_t, t)$. The Jacobian trace is estimated via Hutchinson's stochastic estimator [27].

While our method achieves competitive NLL scores across datasets, we note that direct comparison across models can be influenced by the choice of different priors. In particular, von Mises distributions

with higher concentration can artificially boost log-likelihood scores, whereas broader priors can deflate them. Thus, while our results confirm the effectiveness of our model, qualitative evaluations remain critical for comprehensive assessments.

## B.4 Navier-Stokes Fluid Velocity Field

To demonstrate the applicability to real angular data, we evaluate our Kuramoto diffusion model on 2D incompressible Navier–Stokes (NS) velocity fields from [6]. Each velocity frame $(\boldsymbol{v}_x, \boldsymbol{v}_y)$ is converted to polar form:

$$\boldsymbol{r} = \sqrt{\boldsymbol{v}_x^2 + \boldsymbol{v}_y^2}, \quad \boldsymbol{\theta} = \texttt{arctan2}(\boldsymbol{v}_y, \boldsymbol{v}_x) \tag{20}$$

where $\boldsymbol{r} \geq 0$ is the amplitude and $\boldsymbol{\theta} \in (-\pi, \pi]$ is the phase. Since $\boldsymbol{r}$ is positive, we work in log–magnitude $\boldsymbol{z} = \log \boldsymbol{r}$ and apply a VP SDE there in the log space, while phases evolve via the locally coupled Kuramoto SDE. The forward processes are defined as:

$$\texttt{Amplitude:} \frac{d\boldsymbol{z}_t^i}{dt} = -\frac{1}{2}\beta_t \boldsymbol{z}_t^i + \boxed{\frac{1}{|\mathcal{N}_i|}\sum_{j\in\mathcal{N}_i} K(t)\cos(\boldsymbol{\theta}_t^j - \boldsymbol{\theta}_t^i)} + \sqrt{\beta_t}\xi^i, \quad \boldsymbol{z}_0 = \log \boldsymbol{r}_0$$

$$\texttt{Phase:} \frac{d\boldsymbol{\theta}_t^i}{dt} = \boxed{\boldsymbol{r}_t^i}\left[\frac{1}{|\mathcal{N}_i|}\sum_{j\in\mathcal{N}_i} K(t)\sin(\boldsymbol{\theta}_t^j - \boldsymbol{\theta}_t^i) + K_{\text{ref}}(t)\sin(\psi_{\text{ref}} - \boldsymbol{\theta}_t^i)\right] + \sqrt{2D_t}\xi^i \tag{21}$$

where the boxed terms couple the two channels: (i) local phase coherence accelerates amplitude growth (via cos in the $z$-SDE), and (ii) larger amplitude strengthens phase synchronization (amplitude factor $\boldsymbol{r}_t^i$ in the $\theta$-SDE). To avoid over-coupling late in diffusion (when noise dominates), we enable coupling only in the early stage, *i.e.*, for $t < T/\alpha$ with a constant $\alpha > 1$; for $t \geq T/\alpha$ the two SDEs evolve independently under their native log-VP/Kuramoto dynamics.

In the reverse process we train two score models, one in log–magnitude space and one on phases, to solve the corresponding reverse SDEs and jointly synthesize $(\boldsymbol{r}, \boldsymbol{\theta})$.

**Unconditional Generation.** Following fluid–dynamics practice, we assess realism in the spectral domain at the final time step. From each 2D velocity field, we compute the radial energy spectrum $E(k)$ and then: (i) fit a line to $\log E(k)$ vs. $\log k$ over the mid–wavenumber band $k \in [0.1, 0.4] \cdot k_{max}$ (with $k_{max} = \min(H, W)/2$) and report the absolute slope difference, and (ii) compute the 1D Wasserstein distance between the *normalized* mean spectra of real and generated samples.

Table 9: Spectral evaluation of generated Navier-Stokes fluid velocity fields.

| Metrics | Slope Difference ($\downarrow$) | Wasserstein Distance ($\downarrow$) |
|---|---|---|
| SGM (Amplitude-Phase Decomposition) | 0.7435 | 0.0015 |
| SGM (Cartesian Coordinates) | 0.5590 | 0.0029 |
| Ours (Naive Kuramoto Phase + Log-VP Amplitude) | 0.6954 | 0.0011 |
| Ours (Coupled Kuramoto Phase + Log-VP Amplitude) | **0.3343** | **0.0005** |

Table 9 shows that the Coupled Kuramoto Phase + Log-VP Amplitude model achieves the best spectral realism among all methods, with the lowest slope error and Wasserstein distance. Relative to the SGM (Amplitude–Phase Decomposition) baseline, this corresponds to a $55\%$ reduction in slope error and a $67\%$ reduction in Wasserstein distance. It also improves over SGM (Cartesian Coordinates) by $40\%$ in slope error and $83\%$ in Wasserstein, and over the Naive Kuramoto + Log-VP variant by $52\%$ (slope) and $55\%$ (Wasserstein). These gains indicate that explicitly coupling phase synchronization with amplitude evolution yields more physically plausible velocity spectra than treating angle and magnitude independently or modeling them without coupling. Fig. 10 plots the average energy spectra with fitted lines, where the coupled model shows a visibly tighter fit in the mid-frequency band and a closer overall spectral shape.

**Conditional Forecasting.** Building on the unconditional results above, we next consider a setting that is arguably even more useful in PDE modeling: forecasting future states from history. Unlike unconditional generation which tests realism in distribution, forecasting probes whether the model has learned the dynamics well enough to extrapolate in time.

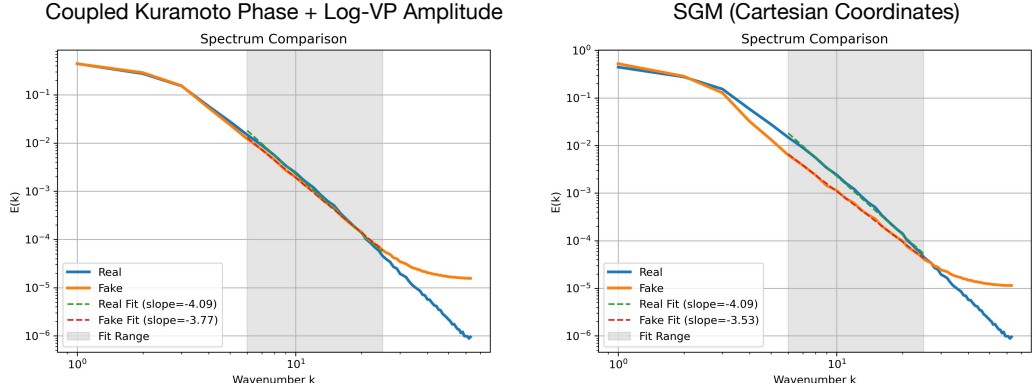

Figure 10: The average energy spectra with fitted lines.

We keep the coupled representations and train two conditioned score models $s_c$ – one for phases $\boldsymbol{\theta}$ and one for log-magnitudes $\boldsymbol{z}$ – together with their unconditioned counterparts $s_u$ via condition dropout. At inference, each pair is fused with classifier-free guidance (CFG) [22]:

$$\begin{aligned}
\texttt{Amplitude:}\, \tilde{s}_\omega(\boldsymbol{z}) &= \boldsymbol{s}_u(\boldsymbol{z}) + \omega\big(\boldsymbol{s}_c(\boldsymbol{z}|c) - \boldsymbol{s}_u(\boldsymbol{z})\big) \\
\texttt{Phase:}\, \tilde{s}_\omega(\boldsymbol{\theta}) &= \boldsymbol{s}_u(\boldsymbol{\theta}) + \omega\big(\boldsymbol{s}_c(\boldsymbol{\theta}|c) - \boldsymbol{s}_u(\boldsymbol{\theta})\big)
\end{aligned} \tag{22}$$

where $c$ denotes the two-step Navier–Stokes history, and $\omega$ is a guidance temperature that controls adherence to the condition (larger $\omega$ denotes stronger conditioning). We evaluate accuracy using Mean Squared Error (MSE) between predicted and ground-truth velocity fields.

Table 10: MSE of Navier-Stokes fluid velocity predictions.

| Methods | MSE ($\downarrow$) |
|---|---|
| SGM (Cartesian Coordinates) | 0.0260 |
| Ours (Coupled Kuramoto Phase + Log-VP Amplitude) | **0.0188** |

As shown in Table 10, the Coupled Kuramoto Phase + Log-VP Amplitude model reduces MSE from 0.0260 to 0.0188 compared with the SGM (Cartesian Coordinates) baseline, indicating more accurate conditional forecasts. Fig. 11 illustrates two examples; predictions closely match the ground truth in both horizontal and vertical velocity components.

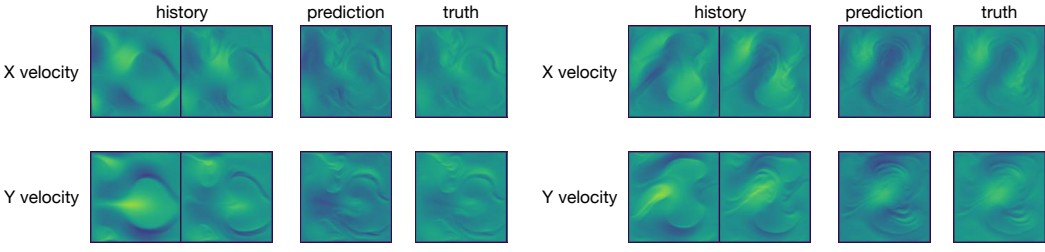

Figure 11: Conditional one-step forecasts of our method on Navier–Stokes velocity: history (left), predictions (middle), and ground truth (right).

## B.5   More Examples of Generative Samples

Figs. 12, 13, 14, and 15 present additional randomly generated samples from our 1000-step locally coupled Kuramoto orientation diffusion model on the SOCOFing fingerprint, Brodatz texture, ground terrain, and CIFAR10 datasets, respectively.

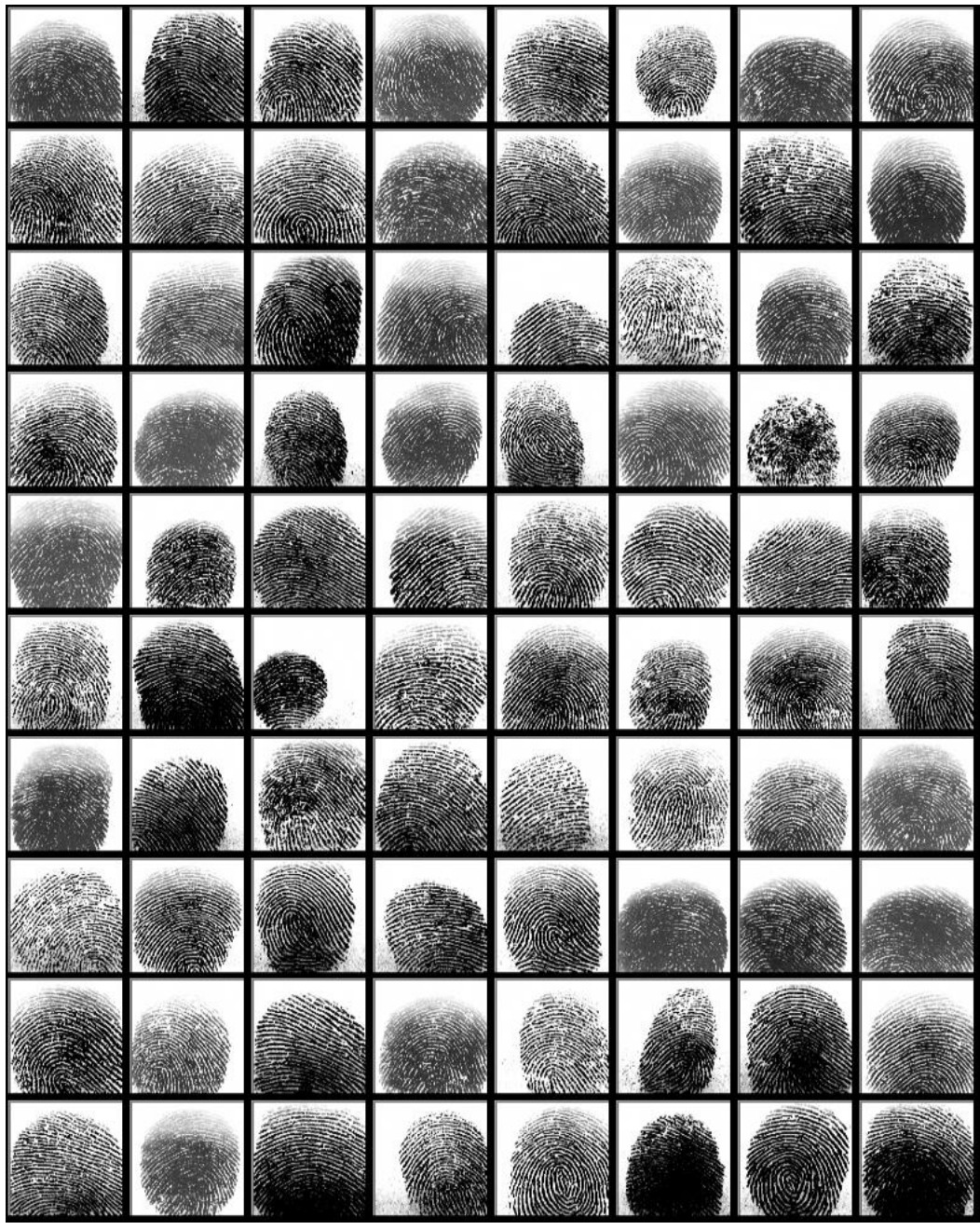

Figure 12: Randomly generated samples on SOCOFing fingerprint dataset.

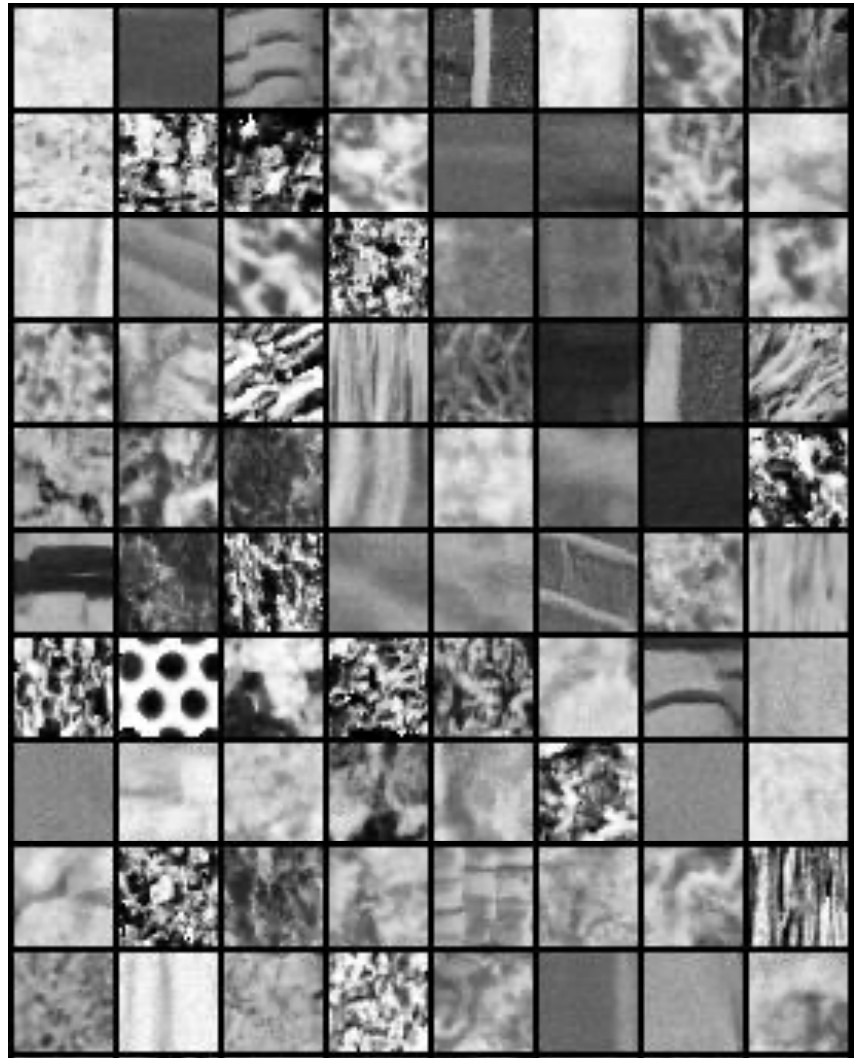

Figure 13: Randomly generated samples on Brodatz textures dataset.

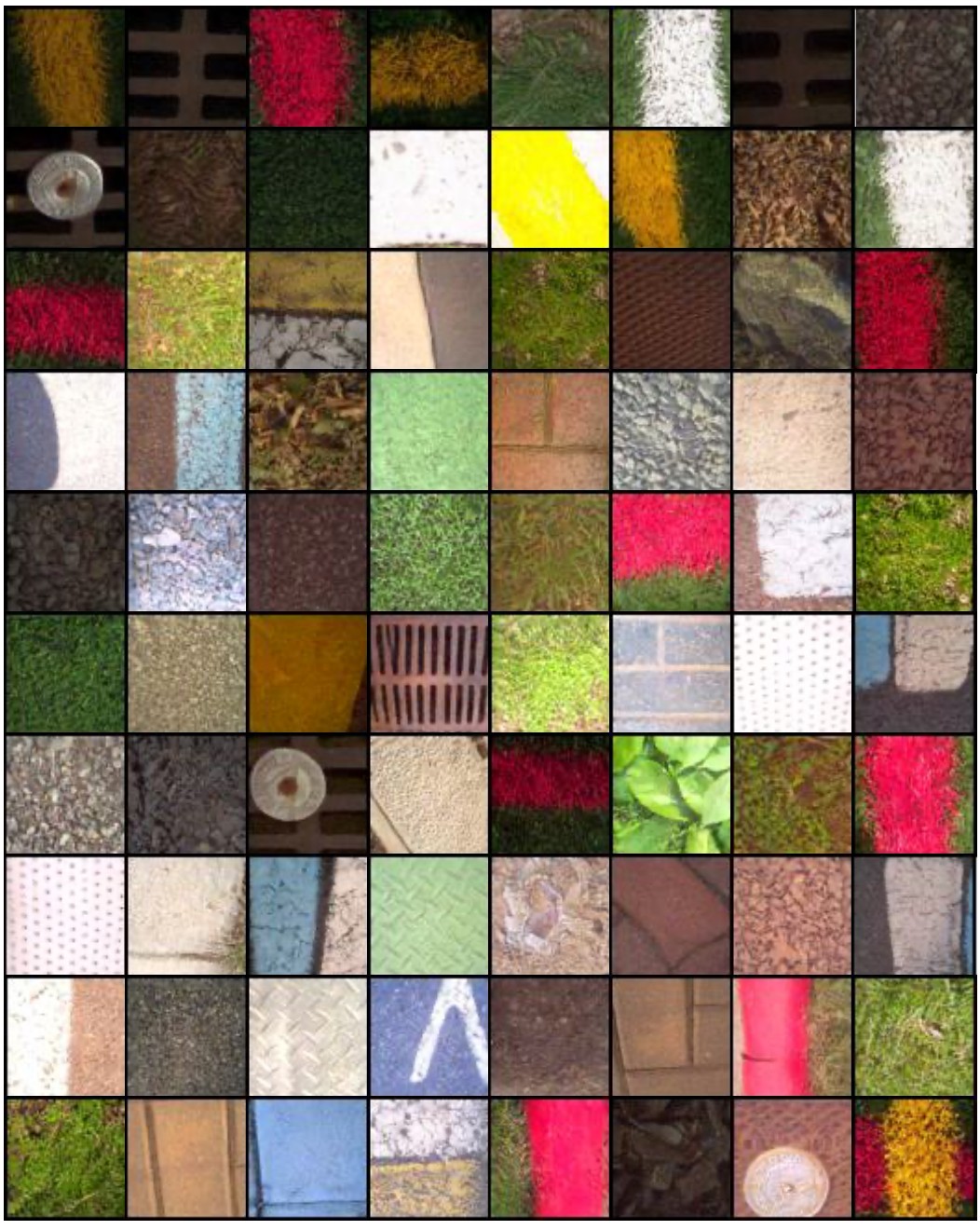

Figure 14: Randomly generated samples on the ground terrain dataset.

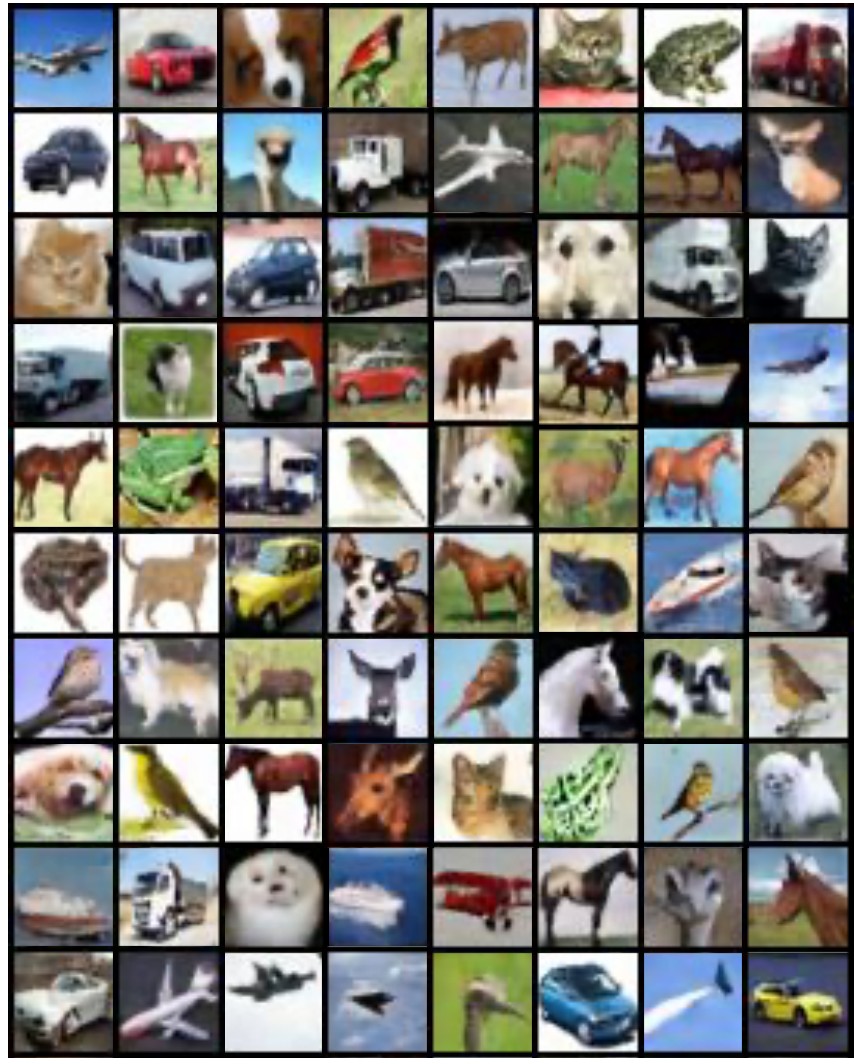

Figure 15: Randomly generated samples on CIFAR10 dataset.

