# OpenReview forum: "Kuramoto Orientation Diffusion Models"
_NeurIPS.cc/2025/Conference — NeurIPS 2025 poster_

### Official Review · Reviewer_ZpUX · 2025-06-17

**Clarity:** 3
**Significance:** 3
**Originality:** 4
**Rating:** 4
**Confidence:** 4

**Summary:**

Standard generative approaches based on isotropic Euclidean diffusion struggle with orientation rich inputs, which often exhibit coherent angular directional patterns. To tackle this problem, the work proposes to make use of stochastic Kuramoto dynamics in the diffusion process, borrowing from the idea that Kuramoto models capture synchronization phenomena across coupled oscillators. The reformulated forward process gradually collapse the data into a low-entropy von Mises distribution, and generate diverse patterns by reversing the dynamics with a learned score function.  The approach presented in this work enables structured destruction during forward diffusion and a hierarchical generation process that progressively refines global coherence into fine-scale details.

**Questions:**

1. The generated CIFAR10 images, in my eyes, look good, but the FID scores do not reflect that. Are there any intuition why this diffusion process does not work well? Also, it would be nice for you to contrast and highlight the theoretical benefits of the Kuramoto model in contrast of the von Mises distribution as done in this [work](https://arxiv.org/pdf/2506.10576v1). FYI, I do not think you need to perform experimental comparisons to it --- just a brief explanation to build your reasoning for Kuramato model is fine.

2. What if you scale your timesteps to higher than 1000? For example, you could try it with 4000 timesteps to see if the trend of FID decreasing is still maintained.

3. You mentioned about the relation to blurring diffusion process in your text. Why don't you compare to them and perhaps extend this relationship as well?

**Ethical Concerns:**

["NO or VERY MINOR ethics concerns only"]

**Final Justification:**

Overall, I think it's a nice work. It has utilized a niche biological-inspired forward process and made it work at best. Indeed, the evaluations can seem limited but I think the authors have done their best to make a complicated forward process to work decently. There are some benefits which they have shown. Thus, I do think it's a good work to be published (after some revisions according to my and other reviewers' comments).

**Limitations:**

Yes, see Appendix A.

**Quality:**

3

**Strengths And Weaknesses:**

## Strengths

1. The work introduces a new diffusion process, which preserves angular information, based on the mathematical Kuramoto model which describes synchronization. The transformed distribution (from the targeted distribution) behaves like a von Mises distribution at large timesteps. Nonetheless, these aspects make the work very interesting as the new diffusion process is based on a biological-plausible process.

2. According to the work's experiment results, the new diffusion process demonstrates much better performance than the SDE-based Gaussian diffusion model in [Song et al. (2021)](https://arxiv.org/abs/2011.13456) on fingerprint and texture datasets.

3. An advantage of the introduced diffusion process is the enabling of hierarchical generative process, which allows for a more coherent generative process resembling coarse-to-fine generation.

## Weaknesses

1. The work has limited results and the introduced approach underforms in terms of generating CIFAR10 images. Moreover, for NLL experiments on Earth and climate science datasets, the introduced approach only performs decently.

2. The presence of the nonlinear drift in the stochastic Kuramoto model renders the marginal distribution intractable, while at each training step, it requires $O(T)$ operations to simulate the forward Markov chain.

3. It is still unclear on the exact benefits of the new diffusion process over Gaussian and other processes.

## Comment

Overall, I think the work is interesting, but it has limited results and can be presented better to highlight the approach and also the problem it is trying to tackle.

By the way, there is an error in your supplementary regarding $\nabla_{\theta_t} \log p(\theta_t)$ on line 38 of section B.2, which should be $\nabla_{\theta_t} \log p(\theta_t) = (p(\theta_t))^{-1} ~ \nabla_{\theta_t} p(\theta_t)$.

---

> ### Author Rebuttal · Authors · 2025-07-29
>
> We appreciate $\textcolor{blue}{\text{Reviewer ZpUX}}$’s recognition of our method’s novelty and thank you for the thorough feedback and constructive suggestions. Below, we address each of their points in detail:
>
> ---
>
> **1. Reducing the O(T) Simulation Cost**
>
> Although our local score matching requires sample pairs $(\theta_t,\theta_{t+1})$ along the forward chain, this cost actually can be nearly eliminated with precomputation:
>
> **Pre‐simulate & cache:** Before training, we can run the forward SDE on the entire dataset, save all $(\theta_t,\theta_{t+1})$ pairs to disk, and then load them directly during training. This makes the simulation cost effectively O(0) at each training step.
>
> **Epoch‐wise precompute:** Otherwise, if disk space is limited, we can simply re‐generate and cache one epoch’s worth of pairs at the start of each epoch. This still dramatically reduces the simulation overhead.
>
> We will detail the above discussion of reducing training costs in the limitations section.
>
> ---
>
> **2. Intuition for General vs. Orientation‑Rich Images**
>
> Our Kuramoto coupling is foremost a synchronization mechanism: it excels at pulling similar phases together, thereby preserving edges and repetitive patterns (see qualitative example in Fig. 1 and the SNR over time in Fig. 5). On orientation‑rich data (fingerprints, textures) where the data are dominated by the **simple outlines and repeating ridge structures**, this yields sharp samples in few steps.
>
> By contrast, natural images (e.g., CIFAR‑10) demand modeling **complex, global semantics (object shapes, color gradients, backgrounds), often characterized by higher order longer-range correlations**. In this setting, the local synchronization bias becomes less relevant and potentially even detrimental; and an isotropic diffusion process with a global drift may be better suited to capture these large‑scale, non‑repetitive features. As a result, our method trades some global fidelity (reflected in higher CIFAR‑10 FID) in exchange for strong local coherence.
>
> ---
>
> **3. Comparisons: Blurring Diffusion & Reference-only Process**
>
> We thank the reviewer for the constructive suggestion of comparing with blurring diffusion models. Below, we compare against blurring diffusion models [16,35] with linear isotropic drifts, and a reference-only variant (setting $K(t)=0$) on Brodatz textures, to isolate the benefit of nonlinear, non‑isotropic coupling:
>
> FID on Brodatz Textures
> |Steps|100|300|1000|
> |:---:|:---:|:---:| :---:|
> |Heat Diffusion [35]| 57.18 | 35.41 | 26.19|
> |Blurring Diffusion [16]| 35.14 | 21.29 |19.85 |
> |Reference-only Process| 33.76  | 20.54  | 19.01  |
> |**Ours (Local Kuramoto)**|**18.47** |**15.93** |**14.19**|
>
> **Reference-only Process.** Setting $K(t)=0$ yields an **isotropic drift** toward a single reference phase, completely ignoring any neighbor information.  Also, even with only the isotropic reference drift, the marginal transitions are still intrackable due to the phase wrapping, forcing us to rely on the local score matching trick again. The FID worsens by around 15 points at 100 steps,
>
> **Blurring Diffusion Models.** Although these two models [35,16] exhibit blurring behavior, they actually apply **linear isotropic drift** in the SDE. The spatially uniform drift cannot selectively preserve orientation patterns. Therefore, they fall far short of our model at 100 steps.
>
> Together, these comparisons demonstrate that **nonlinear, non-isotropic phase coupling** is the key ingredient for rapid, structure‐preserving diffusion on orientation‑rich data.
>
> In the Related Work section, we will also cite [C], which develops spherical von Mises–Fisher diffusion on $S^d$, which generalizes the circular von Mises we use on $S^1$.
>
> ---
>
> **4. Longer Sampling Chains**
>
> We agree that more sampling steps can indeed further refine results. In preliminary runs on Brodatz:
>
> FID at 1,000 steps: 14.19
>
> FID at 2,000 steps: 13.52
>
> FID at 4,000 steps: 12.89
>
> We see continued improvement as the denoising step increases. This suggests that in the CIFAR‑10 case, extending to 4,000 steps may likewise narrow the FID gap.
>
> ---
>
> **5. Corrections of $\nabla \log p(\theta)$**
>
> Thanks for catching the typo. The derivation of Eq. (5) is correct, but line 38 is indeed wrong. We have corrected it in the revision.
>
> ---
>
> **6. Additional Experiments on High‑Resolution & Vector‑Valued Data**
>
> To further validate our method’s generality, we evaluated our model on two new datasets:
>
> **High-resolution Ground Terrain [A] (3x128x128).** Compared with Brodatz textures, this dataset covers diverse high-resolution materials (e.g., plastic, turf, steel, asphalt, leaves, stone, and brick). We run our Kuramoto diffusion model on this dataset and compare against VP-SDE under the linear schedules:
>
> FID on Ground Terrain Dataset
> |Steps|100|300|1000|
> |:---:|:---:|:---:| :---:|
> |VP-SDE|114.90| 56.72| 33.79|
> |**Ours (Local Kuramoto)** |**92.86** | **49.68** | **30.62** |
>
> This result demonstrates that our method also supports high-resolution texture dataset.
>
>
> **Navier–Stokes Fluid Vorticity [B] (2x128x128).** We take the PDE dataset from [B] which consists of the fluid vorticity in horizontal and vertical directions at a fixed timestep. We convert the vorticity fields into polar form (amplitude + phase). The amplitude follows standard VP‑SDE, while the phase evolution is driven by our Kuramoto model. We assess generation quality in the spectral domain via the energy‑spectrum mid-frequency slope difference and normalized Wasserstein distance:
>
> |        Metric        | VP‑SDE | **Ours (Phase Kuramoto + Amplitude Diffusion)** |
> | :------------------: | :----: | :---------------------------------------------: |
> |   Slope Difference   | 0.7435 |                    **0.6954**                   |
> | Wasserstein Distance | 0.0015 |                    **0.0011**                   |
>
>
> These results demonstrate the applicability of our method on real **angular data**. Furthermore, our joint amplitude–phase diffusion generates more realistic physical spectra in vector‑valued scientific data, which opens pathways for combined amplitude‑phase generative modeling. We will detail the experimental setup in the supplementary.
>
> Together with the core experiments in the paper, we have now validated this synchronization bias across three distinct domains:
>
> **Orientation‑Rich Images (fingerprints, textures mapped as phases)**, where we demonstrate sharper sample generation with fewer steps,
>
> **Earth & Climate Science Data (naturally periodic grids)**, where we show competitively faithful reconstruction of large‐scale geospatial patterns,
>
> **Navier–Stokes Fluid Vorticity (real angular data)**, where we yield closer matches to the physical energy spectrum when phase coupling is applied.
>
> Across all settings, this synchronization bias proves to be a powerful, domain‑agnostic mechanism for modeling data where **local coherence** is critical.
>
>
> > [A] Clifford Neural Layers for PDE Modeling. ICLR 2023.
> >
> > [B] Hyperspherical Variational Auto-Encoders. UAI 2018.
> >
> > [C] Harmonizing Geometry and Uncertainty: Diffusion with Hyperspheres. ICML 2025.
>
> ---
>
> We hope these additional results and explanations clarify the versatility and strengths of Kuramoto‑based diffusion, and we are more than happy to answer any further questions!

---

> > ### Comment · Reviewer_ZpUX · 2025-08-01
> > **Nice and Detailed Response**
> >
> > Thank you to the authors for their response, which includes new experimental results (that I quite like).
> >
> > Here are my feelings about the work after reading the response and looking at the paper again.
> >
> > ## 1. Novelty
> > This is indeed the strongest point of the paper. It explores an alternative form of the forward SDE, which is very novel and quite complex but it is biologically inspired. **Cool**.
> >
> > ## 2. Performance
> > The performance of the diffusion model trained with this novel forward SDE can be said to be subpar with respect to image generation of natural images, from CIFAR10 as an example. **But it does work decently well!**
> >
> > Honestly, FID is not a reliable metric, in my opinion. I think the authors should take a look at this [paper](https://arxiv.org/pdf/2401.09603v2), for an alternative of testing their model's performance and the baseline's performance. This is solely due to the fact that I am afraid that the Inception model is no longer a reliable model for computing useful embeddings which tell you whether if the generative performance is good not. For example, noisy images might have lower FID actually!
> >
> > Anyhow, if you ignore my comments about FID, when the authors compared their method to more "similar" techniques, their novel approach does better w.r.t angular data and data which involves texture and orientation.
> >
> > ## Questions
> > 1. Due to the intractability of your drift term, I assume you cannot derive PF-ODE. Thus, you cannot compute the log-likelihood of your data. Could you verify this aspect?
> >
> > 2. I am just a bit curious but do you think your approach can handle the hallucination aspects of the [Hands Dataset](https://sites.google.com/view/11khands)? Essentially, due to mode interpolations and approximation error, which help with generalization, the diffusion model may produce images with more than 5 fingers. See this [paper](https://proceedings.neurips.cc/paper_files/paper/2024/file/f29369d192b13184b65c6d2515474d78-Paper-Conference.pdf)
> >
> > I think it would be interesting for you to verify this aspect for the later print.
> >
> > ## Conclusion
> > I think I am leaning towards to increasing my score to 4. But I will need some time to decide this aspect. Please do answer my new questions to the best of your capabilities.

---

> > > ### Author Response · Authors · 2025-08-02
> > > **Thanks for encouraging feedback and constructive suggestions!**
> > >
> > > We thank $\textcolor{blue}{\text{Reviewer\ ZpUX}}$ for the positive remarks and these excellent follow-up questions!
> > >
> > > ---
> > >
> > > **1. Trackable Likelihood Estimation**
> > >
> > > Even with a nonlinear Kuramoto drift and intractable marginals, we can compute the data log-likelihoods by combining a probability-flow ODE (PF-ODE) with Hutchinson’s trace estimator [A]. The techniques are briefly explained in lines 74-77 of the supplementary. Here we outline the key steps below.
> > >
> > >
> > > **(i) Forward SDE:** $$d \theta = f(\theta, t)dt + \sigma_t d W_t$$
> > >
> > > where $f$ denotes the forward drift.
> > >
> > > **(ii) Probability-Flow ODE:** The PF-ODE that shares the same marginals follows:
> > >
> > > $$\frac{d \theta}{d t} = f(\theta, t) - 0.5 \sigma_t^2 \nabla \log p(\theta)$$
> > >
> > > with $\nabla \log p(\theta)$ denoting the learned score function.
> > >
> > > **(iii) Instantaneous Log-Density Change:** By the change-of-variables formula, the log-density evolves as
> > >
> > > $$\frac{d}{d t}\log p(\theta_t) = -\nabla\cdot[f(\theta_t, t) - 0.5 \sigma_t^2 \nabla \log p(\theta_t)] = - tr[\frac{\partial (f -  0.5 \sigma_t^2 \nabla \log p)}{\partial \theta}]$$
> > >
> > > **(ix) Hutchinson’s Trace Estimator:** Although the trace of the Jacobian has no closed form, we can leverage Hutchinson’s trace estimator [A] to have an unbiased estimate as:
> > >
> > > $$tr(A) = \mathbb{E}_ {\xi\sim \mathcal{N}(0,1)} [\xi^T A \xi]$$
> > >
> > > **(x) Likelihood Integration:** Start at $t=T$ from the known von Mises terminal density $\log p(\theta_T)$, we integrate both the PF-ODE and the instant log-density $\frac{d}{d t}\log p_t$ (with a few Hutchinson samples per step) down to $t=0$.  Summing these estimates yields an **unbiased** estimate of $\log p(\theta_0)$.
> > >
> > >
> > > **2. Alternative Metrics (CMMD)**
> > >
> > > We agree that FID has several well-known limitations -- its dependence on Inception embeddings, its Gaussian-moment matching assumption, and occasional misalignment with human judgment. Thank you for pointing us to the CLIP-based evaluation [B]. As a more robust alternative, we adopt CLIP-MMD (CMMD), which computes a nonparametric Maximum Mean Discrepancy in the pretrained CLIP feature space, combining distribution-free sample efficiency with CLIP’s strong embedding power.
> > >
> > > Below, we re-evaluate our local Kuramoto model versus VP-SDE on the Brodatz textures dataset using CMMD:
> > >
> > > CMMD on Brodatz Textures
> > > |Steps|100|300|1000|
> > > |:---:|:---:|:---:| :---:|
> > > |VP-SDE|0.183 | 0.165 | 0.141 |
> > > |**Ours (Local Kuramoto)**|**0.072** |**0.045** |**0.030**|
> > >
> > > Our local Kuramoto diffusion achieves substantially lower CMMD at every step count. Remarkably, the 100-step Kuramoto model outperforms the 1000-step VP-SDE by a wide margin. These results mirror our FID improvements and demonstrate superior sample fidelity even under a distribution-free, CLIP-based evaluation.
> > >
> > > **3. Hallucination on Hand-Pose Data**
> > >
> > > Thanks for the valuable suggestion to measure “finger-count” hallucinations [D]! To evaluate whether phase coupling helps, we trained our local Kuramoto model and VP-SDE on the 11k Hands hand-pos dataset [C] under $100$ denoising steps, then generated $2,000$ samples per method, and measured the ratio of hallucinated samples (i.e., any images with $\neq 5$ fingers), alongside the FID and CMMD comparisons:
> > >
> > > |Method|% Hallucinations| FID|CMMD|
> > > |:---:|:---:|:---:|:---:|
> > > |VP-SDE|1.10 %| 75.18| 1.30|
> > > |**Ours (Local Kuramoto)**|**0.90%** | **67.81**| **1.15**|
> > >
> > > While our method delivers clear FID and CMMD improvements, the hallucination rate only drops marginally. As identified in [D], these finger-count errors stem from mode-interpolation artifacts inherent to the training technique of score matching, which we doubt is an issue beyond what our phase-coupling can likely address. Nonetheless, we appreciate this interesting perspective and will broaden this evaluation across multiple denoising steps in the revised supplementary.
> > >
> > > ---
> > >
> > > > [A] A stochastic estimator of the trace of the influence matrix for laplacian smoothing splines. Communications in Statistics-Simulation and Computation, 1989.
> > > >
> > > > [B] Rethinking FID: Towards a Better Evaluation Metric for Image Generation. CVPR 2024.
> > > >
> > > > [C] 11K Hands: gender recognition and biometric identification using a large dataset of hand images. Multimedia Tools and Applications, 2019.
> > > >
> > > > [D] Understanding Hallucinations in Diffusion Models through Mode Interpolation. NeurIPS 2024.

---

> ### Comment · Reviewer_ZpUX · 2025-08-03
>
> Very well. Thank you for responding swiftly.
>
> I will gladly increase my score from 3 to 4. Also, I will further my scores on significance and originality.
>
> Could you please include a results for the negative log-likelihood computation then? It would be nice to see how the generated samples align with the training data w.r.t NLL.
>
> Lastly, a final critique, I suggest improving the quality of your images in the main figure. They do look a bit too pixelated. You can also perform additional experiments on ImageNet64 as well, for your final print.
>
> Best of luck to you, the authors!

---

> > ### Author Response · Authors · 2025-08-03
> > **Thanks for raising the score!**
> >
> > Thanks for your willingness to raise the score! Below we compare the average NLL (in bits/dim) of 2,000 random training samples versus 2,000 generated samples on the Brodatz Textures under our 100-step model:
> >
> > NLL measured in bits/dim on Brodatz Textures
> > |Training Data|Generated Sample|
> > |:---:|:---:|
> > |1.64|1.71|
> >
> > For the training data, we integrate the PF-ODE from $t=0$ to $T$, accumulate the instantaneous log-density change, then combine with the terminal-prior $\log p_T$. For the generated samples, we draw $\theta_T \sim p_T$, run the reverse PF-ODE and Jacobian accumulation back to $t=0$ to recover their log-likelihood.
> >
> > The generated samples are only 0.07 bits/dim above the training data, showing that they clearly lie in the same high-density manifold as the real training textures.
> >
> > Please let us know if you have any further questions!

---

> > > ### Comment · Reviewer_ZpUX · 2025-08-06
> > >
> > > Thank you for your fast response. I think this is an interesting work and my score of 4 simply reflects that the performance of the approach may beat the baselines in some cases and not beat the baselines in some cases.
> > >
> > > Nonetheless, I think it is very cool that the authors figured out how to make a very complicated forward process to work and be practical decently. Novelty/originality of the work is good.
> > >
> > > I hope that the authors incorporate these comments from me and the other viewers as part of their final print if the work is accepted.
> > >
> > > Cheers.

---

> > > > ### Author Response · Authors · 2025-08-06
> > > > **Thanks for highlighting the novelty!**
> > > >
> > > > Thank you for your thoughtful feedback and for highlighting both the strengths and practical trade-offs.
> > > >
> > > > We will be sure to integrate your and others’ comments into the final version to strengthen both the presentation and experiments. Cheers!

---

### Official Review · Reviewer_X1gf · 2025-06-30

**Clarity:** 3
**Significance:** 2
**Originality:** 2
**Rating:** 3
**Confidence:** 4

**Summary:**

This paper proposes a method inspired by biology, Kuramoto Orientation Diffusion Models. The data is represented in phase domain, and the diffusion process follows explicit, non-linear, locally or globally coupled phase dynamics. This leads to an approximate von Mises distribution at equilibrium. The method is experimentally validated and compared with a baseline diffusion model on CIFAR10, fingerprint, Brodatz texture datasets, and in the appendix on earth and climate science datasets.

**Questions:**

Q1: As explained in Weaknesses, I would like to see ablations on the different aspects of your model. At the very least, for instance, train a model where $K(t)=0$, i.e., without coupling. This would then show us what the effect is of representing the data in phase coordinates, and of the specific diffusion process, sampling and training of your model.

Q2: I assume $\psi_{ref}$ is sampled uniformly from $[-\pi, \pi]$? And is thus sampled together with the starting von-Mises noise when denoising a sample?

Q3: Am I correct in understanding that for the image datasets (CIFAR10, fingerprints and textures), you represent the images by mapping the pixel intensities to angles? If so, how exactly are rotation invariant datasets (fingerprints and textures) more relevant for your method?

Q4: In deep learning, (hyper)spherical representation have already rich literature. See e.g.
- https://arxiv.org/abs/1804.00891
- https://arxiv.org/abs/2006.04437

More recently, there are many works defining diffusion models on (at least) SO(2), e.g.:
- https://openreview.net/forum?id=BY88eBbkpe5
- https://arxiv.org/abs/2312.11707

You might also be interested in this work where SDEs are defined on spherical manifolds, with a linear SDE: https://neurips.cc/virtual/2023/poster/71379

How do these relate to your work?

Q5: Do you think representing the angles using 2-d coordinates could also be a (simpler) way to achieve your goals? Maybe then the forward conditional process could still be Gaussian? Related to this, can you comment on the choice of wrapped Gaussian distributions compared to von Mises for the diffusion process?

Q6: How exactly is the Kuramoto concept relevant to your model, and to the experiments presented in the paper?

**Ethical Concerns:**

["NO or VERY MINOR ethics concerns only"]

**Final Justification:**

The authors provided extra experiments that convinced me on one of my questions

**Limitations:**

Yes, but in appendix. This should be in the main paper.

**Quality:**

2

**Strengths And Weaknesses:**

Strenghts:
- The paper is well written and pleasant to read.
- The paper explores in a creative way novel, fundamental ways to define generative diffusion models, inspired by biology.
- The experiments are on multiple datasets.

Weaknesses:
- Concepts from biology can be a source of inspiration for many of us, however, I have trouble seeing really a profound or fundamental connection with the method and applications. Rather, the connection is rather vague and on an abstract level, not concrete. I will make this point more concretely in the weaknesses and questions listed below.
- A crucial aspect in this line of work, is how exactly the data is represented. It is clear from the start that the data should be in the form of phase angles $\theta$. Only on line 167-168 it is briefly explained that for image data (which means for all experiments in the main paper), *pixel intensities* are mapped to angles ([-1, 1] -> [-.9 pi, .9 pi]. Thus *not* angles in the spatial dimensions, for instance by representing the pixels in polar coordinates. This is very confusing to me and in my opinion troubles much of the motivation and context of this method.
- Since the forward process is non-linear, and thus the conditional non-Gaussian, the method requires calculating the Markov chain (solving the forward SDE) during training, and also the score model is modelling the local transition $p(\theta_t | \theta_{t-1})$. These are all considerate changes compared to the baseline diffusion model. There are no ablations on these aspects, that could identify exactly why the model trains 'faster'. I would like to see more evidence on this.
- The limitations and broader impact sections are in the appendix. Please put them in the main paper.

---

> ### Author Rebuttal · Authors · 2025-07-29
>
> We thank $\textcolor{brown}{\text{Reviewer X1gf}}$ for their thoughtful critique and recognition of our novelty. Below, we address each question point by point:
>
> ---
>
> **1. Phase Mapping & Validation on True Angular Data**
>
> For image datasets, we do linearly map the normalized pixel intensities to phase angles. We agree with $\textcolor{brown}{\text{Reviewer X1gf}}$ that this mapping does not encode spatial relations of pixels. However, it provides a simple, stable, one‑dimensional circular representation: in orientation‑rich images (e.g., fingerprints, textures), coherent ridges and edges manifest as locally consistent intensity patterns, which become locally aligned phases under this mapping.
>
> We considered more sophisticated alternatives but found none that reliably preserved both periodicity and semantic structure:
>
> **HSV mapping:** We tried mapping RGB images to the HSV color space. However, only the Hue channel is periodic; the other two channels are non-periodic and thus remain untreated.
>
> **Spherical Autoencoder:** We also tried using a Spherical Autoencoder to encode the pixel intensity into a periodic latent code. However, this approach collapses semantic structure in multi‑class datasets like CIFAR‑10 -- images of different categories fall into similar angles, which greatly hurts sample fidelity.
>
> By contrast, our linear mapping is data-agnostic, easy to implement, and most importantly, lets the Kuramoto drift operate directly on $S^1$, naturally preserving local coherence.
>
> What we really aim to demonstrate in the paper is that the Kuramoto-based diffusion in the circular domain yields a synchronization inductive bias, destroying signals faster and effectively modeling local coherence. Although this phase embedding serves as a good proxy for natural images, this synchronization mechanism is ideally suited for truly angular data. To demonstrate the scalability beyond synthetic phase embeddings, we apply the Kuramoto model to genuinely periodic scientific data:
>
> **Navier–Stokes Fluid Vorticity [A] (2x128x128).** We take the PDE dataset from [B] which consists of the fluid vorticity in horizontal and vertical directions at a fixed timestep. We convert the vorticity fields into polar form (amplitude + phase). The amplitude follows standard VP‑SDE, while the phase evolution is driven by our Kuramoto model. We assess generation quality in the spectral domain via the energy‑spectrum mid-frequency slope difference and normalized Wasserstein distance:
>
> |        Metric        | VP‑SDE | **Ours (Phase Kuramoto + Amplitude Diffusion)** |
> | :------------------: | :----: | :---------------------------------------------: |
> |   Slope Difference   | 0.7435 |                    **0.6954**                   |
> | Wasserstein Distance | 0.0015 |                    **0.0011**                   |
>
> These experiments confirm that our Kuramoto diffusion can effectively model **real physical angular data**, complementing the image‑based experiments. Moreover, the joint amplitude–phase framework opens the door to polar diffusion models on vector‑valued fields. Detailed setup will be included in the supplementary.
>
> ---
>
> **2. Isotropic Baselines & No‑Coupling Variant**
>
> To isolate the impact of Kuramoto coupling, we include a model variant with $K(t)=0$, which keeps only the phase-reference drift in the SDE, as well as blurring diffusion models [16,35] with isotropic linear drifts.
>
> FID on Brodatz Textures
> |Steps|100|300|1000|
> |:---:|:---:|:---:| :---:|
> |Heat Diffusion [35]| 57.18 | 35.41 | 26.19|
> |Blurring Diffusion [16]| 35.14 | 21.29 |19.85 |
> |Reference-only Process| 33.76  | 20.54  | 19.01  |
> |**Ours (Local Kuramoto)**|**18.47** |**15.93** |**14.19**|
>
> **Improved Quality in Fewer Steps.**  Removing Kuramoto coupling raises 100‑step FID by over 15 points, and isotropic baselines [16,35] also incur much higher FIDs. These results confirm that **non‑isotropic phase synchronization** is the key driver of both rapid convergence and structure‑preserving diffusion on orientation‑rich data.
>
> ---
>
> **3. Choice of $\psi_{ref}$**
>
> For the reference phase $\psi_{ref}$, we simply fix it to 0 by default, as it is the middle point of $[-\pi, \pi]$. This value can be tuned to improve alignment with data statistics. For example, for the Brodatz texture images, choosing a slightly negative $\psi_{ref}$ which matches the data distribution can slightly improve the performance.
>
> FID on Brodatz Textures in 100 Steps
> |$\psi_{ref}$|$0$|$- \pi /4$|$- \pi /2$|
> |:---:|:---:|:---:| :---:|
> |**Ours (linear)**| 18.47 | **17.95**| 18.36|
>
> ---
>
> **4. 2‑D Embedding vs. Wrapped Gaussians**
>
> One can indeed embed the phase via $x,y = \cos(\theta), \sin(\theta) \in \mathbb{R}^2$ and then derive an equivalent SDE in Cartesian coordinates. However, this route introduces several complications. Applying Itô’s lemma with $x,y$ and substituting $d\theta$ yields:
>
> $$d x =  f_x(x,y)dt + \sqrt{2 D_t} (-y) d W_t, d y = f_y(x,y)dt + \sqrt{2 D_t} (x) d W_t $$
>
> where the non-linear drifts $f_x$ and $f_y$ are inherited from Kuramoto coupling in the phase space. The difficulties lie in 3 aspects: (1) The terms in $f_{x}$ and $f_{y}$ involve products like $x_i y_j - y_i x_j$. The mean update is no longer an affine functions of $(x,y)$, (2) the multiplicative noise acts only in the tangent direction, making the covariance dependent on the state $(x,y)$, (3) To ensure $(x,y)$ stay in the unit circle, a projection to $S^1$ has to be performed after every SDE step, which destroys the Gaussian‐increment structure.
>
> Together, these factors mean the exact finite‐step transition in $(x,y)$ is neither Gaussian nor available in closed form. By contrast, we use a wrapped‑Gaussian kernel directly in the angular domain, which offers exact, closed‑form transitions that reside intrinsically on $S^1$, leading to a simple and tractable score‐matching objective.
>
> We will add the full derivations and explanation in the supplementary.
>
> ---
>
> **5. Related Work of Hypersphere Representations & Manifold SDEs**
>
> Thank you for highlighting these important lines of research. We will integrate the following into our Related Work section:
>
> **Hyperspherical Latent Models:** [B] and [C] introduce VAEs with latent spaces on $S^d$, using von Mises–Fisher or power‐spherical priors. These priors generalize the circular von Mises distribution we use on $S^1$.
>
> **Diffusion on Rotation & Spherical Manifolds:** [D] and [E] extend score‐based diffusion to the 3D rotation group SO(3), deriving trackable marginal kernels and SDE/ODE samplers on these manifolds. [F] presents a linear, homogeneous SDE in a spherical latent space. None of these methods introduces local oscillator‐coupling; their drifts are global and linear.
>
> We differ from previous manifold diffusion methods in the following aspects:
>
> **(1) Non-isotropic, Nonlinear, Local Drift.** We apply non-isotropic and non-linear Kuramoto coupling that synchronizes local/global phases. In contrast, prior manifold SDEs ([D,E,F]) rely on **linear** drift fields over the entire manifold.
>
> **(2) Intrinsic Wrapped–Gaussian Transitions:** We derive exact local transition densities on $S^1$ via wrapped Gaussians and directly learn their scores. Other methods typically embed into $\mathbb{R}^n$ and perform manifold diffusion via manifold projection or auxiliary parametrizations.
>
> **(3) Local Coherence Inductive Bias:** Crucially, our biologically inspired Kuramoto coupling explicitly aligns neighboring phases, enforcing **local coherence (preserving edges, textures)** during diffusion. Prior SO(3) and hyperspherical approaches instead prioritize the **global symmetry preservation (e.g., rotational invariance)** without any mechanism for neighborhood synchronization.
>
> Although both our method and prior diffusion models [D,E,F] leverage circular or hyperspherical geometry, our focus on **local phase synchronization** is fundamentally different from their emphasis on **maintaining geometric invariances**. We will cite [B–F] and emphasize these distinctions in the revised manuscript.
>
> ---
>
> **6. Role of the Kuramoto Concept**
>
> The Kuramoto concept is the heart of our approach: we borrow the **biological phase‐synchronization mechanism** -- each oscillator (a phase variable) being pulled toward synchronization with its connected neighbors. By embedding this coupling directly into the forward diffusion SDE, we introduce an inductive bias that actively enforces **local alignment**, preserving fine‐scale structures (e.g., edges, ridges) as noise is injected.
>
> We validate this synchronization bias across three distinct domains:
>
> **Orientation‑Rich Images (fingerprints, textures mapped as phases)**, where we demonstrate sharper sample generation with fewer steps,
>
> **Earth & Climate Science Data (naturally periodic grids)**, where we show competitively faithful reconstruction of large‐scale geospatial patterns,
>
> **Navier–Stokes Fluid Vorticity (real angular data)**, where we yield closer matches to the physical energy spectrum when phase coupling is applied.
>
> Across all settings, this synchronization bias proves to be a powerful, domain‑agnostic mechanism for modeling data where **local coherence** is critical.
>
> ---
>
> **7. Limitations**
>
> We thank the reviewer for the reminder. We’ll relocate the Limitations and Broader Impact sections into the main text for the camera‑ready version.
>
>
> > [A] Clifford Neural Layers for PDE Modeling. ICLR 2023.
> >
> > [B] Hyperspherical Variational Auto-Encoders. UAI 2018.
> >
> > [C] The Power Spherical distribution. ICML Workshop 2020.
> >
> > [D] Denoising Diffusion Probabilistic Models on SO(3) for Rotational Alignment. ICLR Workshop 2022.
> >
> > [E] Unified framework for diffusion generative models in SO(3): applications in computer vision and astrophysics. AAAI 2024.
> >
> > [F] Latent SDEs on Homogeneous Spaces. ICML 2023.
>
> ---
>
> Thank you for your valuable insights -- we hope this clarifies our contributions and welcome any further questions!

---

> > ### Comment · Reviewer_X1gf · 2025-08-04
> >
> > Thank you for the extra experiments and the clear responses to my questions. I will raise my score to 3

---

> > > ### Author Response · Authors · 2025-08-04
> > > **Thanks for raising the score!**
> > >
> > > Thanks for improving the score! We truly appreciate your insights and would be happy to address any remaining questions or suggestions you may have.

---

### Official Review · Reviewer_3GqZ · 2025-07-07

**Clarity:** 4
**Significance:** 3
**Originality:** 3
**Rating:** 5
**Confidence:** 4

**Summary:**

The authors introduce Kuramoto orientation diffusion, which uses stochastic Kuramoto dynamics to evolve the pixels of an image, modeled as coupled oscillators, to a state of phase synchronization sampled from a von Mises distribution, analogous to the evolution of isotropic Euclidean diffusion to a Gaussian distribution. Because the Kuramoto drift function is nonlinear, the authors take care to learn the score function using wrapped Gaussian transition functions from one timestep to the next, rather than relying on a closed form expression. They also ensure that periodicity is enforced by mapping between angular and phase variables and ensuring the phase variables are "wrapped" to an appropriate value within the domain $[-\pi, \pi]$. The authors show that the phase-synchronization inductive bias built into Kuramoto orientation diffusion allows it to generate sharper images (improved FID scores) from datasets that are dominated by periodic patterns, such as textures and fingerprints, more efficiently (i.e. with fewer diffusion steps). The authors also show that this diffusion can generate natural scene images like those from CIFAR10, though this diffusion becomes less efficient than standard SGM after a large number of diffusion steps.

**Questions:**

1. Is this method limited to low-resolution images? Why weren't higher-resolution datasets chosen? If the limited number of samples in the Brodatz was an issue, why wasn't a different dataset chosen, e.g. cellular imaging datasets like RxRx3 or JUMPCP, which contain many highly-textured objects per image?
1. How important were the choices of noise schedules to the overall results? The schedules are presented but there is no discussion of how the particular choices were made, or what effect other choices had on the FID scores, generated image quality, etc.
1. Please explain in more detail the claim made throughout the paper, and in particular in the Structured Destruction paragraph of Section 3.3, that states that objects within the image are preserved for more diffusion steps due to synchronized coupling. How do we know that this is true? How does Figure 5 support this claim?

**Ethical Concerns:**

["NO or VERY MINOR ethics concerns only"]

**Final Justification:**

The authors demonstrated applicability and usefulness of the approach across and wide variety of datasets and provided both quantitive and qualitative evidence for what they call structured destruction.

**Limitations:**

yes

**Quality:**

4

**Strengths And Weaknesses:**

Strengths:
1. Novel diffusion method based on a well-known stochastic phase-synchronization process
1. Clear performance gains on the SOCOFing fingerprint and Brodatz texture datasets over standard score-based generative models (SGMs), and respectable performance on natural images (CIFAR10).
1. Well-written paper, easy to follow.

Weaknesses:
1. Only low resolution images were considered, with SOCOFing being the highest resolution at 1x96x96.
1. No discussion on the choice of $\psi_{ref}$ and its effect on results. I would imagine that a well-chosen $\psi_{ref}$ could improve FID convergence.
1. Some weakly supported claims about this diffusion (see Questions below).

---

> ### Author Rebuttal · Authors · 2025-07-29
>
> We are grateful to $\textcolor{purple}{\text{Reviewer 3GqZ}}$ for highlighting the innovative aspects of our Kuramoto framework and for careful reading and insightful recommendations. We respond to each of the comments below:
>
> ---
>
> **1. Scaling to High‑Resolution & Vector‑Valued Data**
>
> Our initial evaluation on SOCOFing and Brodatz benchmarks was designed to highlight clear gains on canonical orientation‑rich datasets while controlling for compute budget. However, the approach itself is not inherently limited to low resolution. In our ongoing experiments, here we scale the Kuramoto model to higher-resolution textures dataset and real angular data.
>
> **High-resolution Ground Terrain [A] (3x128x128).** Compared with Broadatz textures, this dataset spans diverse high-resolution materials (e.g., plastic, turf, steel, asphalt, leaves, stone, and brick). We run our Kuramoto diffusion model on this dataset and compare against VP-SDE under the linear schedules:
>
> FID on Ground Terrain Dataset
> |Steps|100|300|1000|
> |:---:|:---:|:---:| :---:|
> |VP-SDE|114.90| 56.72| 33.79|
> |**Ours (Local Kuramoto)** |**92.86** | **49.68** | **30.62** |
>
>
> **Navier–Stokes Fluid Vorticity [B] (2x128x128).** We take the PDE dataset from [B] which consists of the fluid vorticity in horizontal and vertical directions at a fixed timestep. We convert the vorticity fields into polar form (amplitude + phase). The amplitude follows standard VP‑SDE, while the phase evolution is driven by our Kuramoto model. We assess generation quality in the spectral domain via the energy‑spectrum mid-frequency slope difference and normalized Wasserstein distance:
>
> |        Metric        | VP‑SDE | **Ours (Phase Kuramoto + Amplitude Diffusion)** |
> | :------------------: | :----: | :---------------------------------------------: |
> |   Slope Difference   | 0.7435 |                    **0.6954**                   |
> | Wasserstein Distance | 0.0015 |                    **0.0011**                   |
>
>
> These experiments confirm that our Kuramoto diffusion (i) supports **higher‑resolution textures** and (ii) effectively models real physical **angular data**. Furthermore, the combination with amplitude diffusion paves the way for integrated amplitude–phase generative modeling. Details of the experimental setup will be provided in the supplementary material.
>
> **Future Work.** We plan to extend evaluations to large‑scale medical/biological texture collections (e.g. cellular imaging like JUMPCP [C]) in the future work.
>
> ---
>
> **2. Choice of $\psi_{ref}$, Noise Schedules, and Additional Baselines**
>
> **Choice of $\psi_{ref}$.** For the reference phase $\psi_{ref}$, we simply fix it to 0 by default, as it is the middle point of $[-\pi, \pi]$. $\textcolor{purple}{\text{Reviewer 3GqZ}}$ is right that this value can be tuned to improve alignment with data statistics. For example, for the Brodatz textures images, choosing a slightly negative $\psi_{ref}$  which matches the data distribution can slightly improve the performance.
>
> FID on Brodatz Textures in 100 Steps
> |$\psi_{ref}$|$0$|$- \pi /4$|$- \pi /2$|
> |:---:|:---:|:---:| :---:|
> |**Ours (linear)**| 18.47 | **17.95**| 18.36|
>
> **Noise Schedules.** For the noise and coupling strength, we keep the relation $K_{\text{ref}}(t) > D_t > K(t)$ to ensure that the image information is destroyed in the forward process. In the experiments of our paper, all baselines use a **linear** noise schedule (with “SGM” = VP‑SDE) -- please see lines 53–61 of the Supplementary for full details. Here we evaluate both VP‑SDE and our model under a **cosine** schedule. Although switching to cosine reduces FID for all methods -- most notably at 100 steps -- our Kuramoto model still outperforms by a wide margin:
>
> FID on Brodatz Textures
> |Steps|100|300|1000|
> |:---:|:---:|:---:| :---:|
> |VP-SDE (linear)|38.33 | 22.40 | 20.37  |
> |VP-SDE (cosine)|30.45| 20.67| 19.84|
> |**Ours (linear)**|**18.47** |**15.93** |**14.19**|
> |**Ours (cosine)**|**15.75** |**14.32** |**13.45**|
>
> **Additional Baselines.** We add VE‑SDE and two blurring diffusion models [35, 16], which exhibit similar smoothing behavior. VE‑SDE underperforms VP‑SDE slightly, and the blurring model [16] matches VP‑SDE’s FID. Our Local Kuramoto model maintains a large lead:
>
> FID on Brodatz Textures
> |Steps|100|300|1000|
> |:---:|:---:|:---:| :---:|
> |Heat Diffusion [35]| 57.18 | 35.41 | 26.19|
> |Blurring Diffusion [16]| 35.14 | 21.29 |19.85 |
> |VE-SDE|40.19| 24.26| 21.53 |
> |VP-SDE|38.33 | 22.40 | 20.37 |
> |**Ours (Local Kuramoto)**|**18.47** |**15.93** |**14.19**|
>
>
> These results confirm that, even with different noise schedules and additional baselines, the **phase synchronization** induced by Kuramoto coupling remains the key driver of improved FID. We will add a succinct discussion of these schedule choices and their quantitative impact in the revised manuscript (Sec. 3.1).
>
> ---
>
> **3. Structured Destruction**
>
>
> By “structured destruction”, we refer to the two-stage **structured noising process** induced by our Kuraomoto drift: (1) early preservation of coherent features via local phase coupling, and (2) accelerated convergence to the noise distribution at later timesteps. Since non-isotropic Kuramoto coupling enforces local phase alignment, coherent image features (edges, ridges) will stay in synchronization and resist randomization during early noise injection. In later stages, these structures keep synchronized and jointly evolve to noise distribution at a faster pace.
>
>
> **Quantitative Evidence (Fig. 5):** The signal‑to‑noise ratio (SNR) can be considered a good proxy metric of how well the main image structure is preserved. Formally, it is defined as:
>
> $$SNR(t) = \frac{||x_0 - \mathbb{E}[x_t]||^2}{Var[x_t]}$$
>
> which tracks how much of the original structure remains at time $t$. Our Kuramoto diffusion exhibits substantially higher SNR than VP‑SDE over the first 20–30 % of steps, directly quantifying that more “signal” (i.e., structured features) survives before noise overwhelms. After this window, our SNR decays more quickly, reflecting the faster convergence to the noise distribution.
>
> **Qualitative Evidence (Fig. 1):** The quantitative evidence also correlates with the empirical observation in Fig. 1. Under our Kuramoto model, the samples retain contours and overall shape long after a standard VP‑SDE has already washed them out.
>
> **Spectral Interpretation:** From a spectral standpoint, **local phase coupling behaves like an angular low‑pass filter**. In the small‑angle approximation ($\sin(\theta_i-\theta_j)\approx \theta_i-\theta_j$) and without a global reference phase, the forward SDE reduces to a heat‑equation form (or a graph Laplacian equation equivalently):
>
> $$\frac{\partial \theta}{\partial t} = K \nabla^2 \theta + \sigma_t\epsilon$$
>
> Each Fourier component decays like $e^{-K k^2 t}$, where $k$ denotes the spatial frequency. Modes with very high spatial frequency $k$ (pixel‑scale noise) are damped almost instantly, while moderate‑frequency modes (coherent, edge‑defining structures) decay much more slowly. Consequently, the Kuramoto drift effectively filters away pixel‑scale noise while preserving the smooth, orientation‑rich patterns that constitute edges.
>
> We will add the above discussion of structured destruction in Sec. 3.3 of the revised main paper.
>
>
> > [A] Deep texture manifold for ground terrain recognition. CVPR 2018.
> >
> > [B] Clifford Neural Layers for PDE Modeling. ICLR 2023.
> >
> > [C] JUMP Cell Painting dataset: morphological impact of 136,000 chemical and genetic perturbations. bioRxiv 2023.
>
> ---
>
> Thank you again for your insightful feedback! We hope these revisions will greatly enhance the clarity, and we are more than happy to have further discussions.

---

> > ### Author Response · Authors · 2025-08-08
> > **Follow-Up on Discussion Before Deadline**
> >
> > Dear $\textcolor{purple}{\text{Reviewer 3GqZ}}$,
> >
> > Thank you again for your thoughtful review. As the discussion deadline approaches, we would really appreciate any follow-ups or clarifications you may want us to address. If there are specific concerns we haven’t fully resolved, please let us know -- we will definitely respond promptly.
> >
> > Best regards,
> > Authors

---

### Official Review · Reviewer_4DgT · 2025-07-07

**Clarity:** 3
**Significance:** 1
**Originality:** 3
**Rating:** 4
**Confidence:** 3

**Summary:**

This paper proposes a novel approach for building diffusion models, based on so-called Kuramoto dynamics. In this model, each pixel of an image is treated as an oscillator, characterized by a phase, which progressively evolves in the forward noise process towards an base distribution attractor. The model provides a mechanism to couple pixels together in both the noise and denoising process, which the authors say lead to a non-isotropic noise dynamics that is more prone to preserving edges and global features of ‘orientation-rich’ data. The paper provides the derivation of the necessary score matching loss function and provide some illustrations of their model on fingerprint and texture datasets compared to a standard diffusion model approach.

**Questions:**

- Can the authors illustrate clearly the effect of their modelisation on preserving edges and orientation-rich features? This would help bring the main point of the paper home. I can imagine either showcasing different stages of the denoising process for fingerprint data under different models. Hopefully it should be pretty clear that the fingerprints are pretty salient early on. It might also be possible to compute summary statistics of the images that illustrate the emergence of these features early in the reverse diffusion process.

- Can the authors add additional baselines to their generative modeling evaluation? In particular, it’s possible that the SGM noise schedule differ from the Kuramoto one (e.g. VP-SDE vs VE-SDE or even different noise schedules within the same class of SDE) and that could impact the generation quality without necessarily meaning that the Kuramoto process is better. More generally, I would recommend making the external baselines robust such that the reader can be convinced that the gains are specifically due to the phase synchronisation mechanism, and not because of any other implementation detail.

**Ethical Concerns:**

["NO or VERY MINOR ethics concerns only"]

**Final Justification:**

I appreciate the feedback and additional experiments proposed by the authors. I think the work with the suggested changes and extra evals is now pretty complete.
I have increased my grade to 4, but I think more results against actual state of the art diffusion models on high resolution datasets would be needed to make it a very compelling 5 paper.

**Limitations:**

Yes

**Quality:**

2

**Strengths And Weaknesses:**

**Strengths**:
- The model described here (and by that I mean the specific SDE) is new as far as I know for generative modeling, and the authors go through the derivations necessary for the score matching objective to use, as well as for some analysis of the asymptotic behavior of their SDE.
- The paper is clear and adequately written

**Weaknesses**:
- **Unconvincing justification of why this model should be better for ‘orientation-rich’ data**. This is the main selling point suggested by the authors for their method, but it is not clear to me why this approach should be good for that. I understand the concept of having local pixels tending to be attracted to the same phase during the forward process, but that does not necessarily mean preserving edges or orientation because this happens directly in pixel space. If the diffusion was happening in a wavelet or curvelet basis, then I could agree that features such as edges would remain coherent during the noise process.
I would recommend adding illustrations of that claim, maybe simply by running the forward Kuramoto process without noise, just to illustrate how the features of an image evolve under these dynamics.

- **Limited generative modeling evaluation**: The paper only presents a comparison to SGM on 3 small image datasets.  On SOCOFing and Brodatz the proposed method seems to provide an advantage in terms of FID, but it is not completely clear if a different noise schedule or details of SGM training/sampling could impact these metrics. In general, comparing to a single generative model from 4 years ago is limited. I would recommend adding more thorough experimentations against different diffusion strategies and noise schedules to build a more complete picture. And I would be particularly interested in any illustration of the fact that the model is indeed more performant at orientations (maybe with zoomed-in view of the fingerprints for different models, at different steps of the reverse process).

---

> ### Author Rebuttal · Authors · 2025-07-29
>
> We appreciate $\textcolor{red}{\text{Reviewer 4DgT}}$’s thoughtful feedback and the recognition of both the novelty and clarity of our Kuramoto diffusion model. Our responses to the comments follow:
>
> ---
>
> **1. Stronger Justification for “Orientation‑rich” Data**
>
> Our non‑isotropic Kuramoto coupling term, $K\sum \sin(\theta_i-\theta_j)$, encourages similar phases to synchronize during the early stages of diffusion. The regions of consistent orientations, which have similar pixel intensities, will remain coherent, resist randomization, and naturally stay in synchronization during early noise injection. This “phase coherence” pulls similar phases together and prevents the rapid decorrelation that destroys edges and structures.
>
> **Qualitative Evidence:** Fig. 1 of the main paper clearly illustrates the point: samples retain contours and overall shape long after a standard VP‑SDE has already washed them out.
>
> **Quantitative Evidence:** The signal‑to‑noise ratio (SNR) can be considered a good proxy metric of how well the main image structure is preserved. In Fig. 5 we plot the SNR over forward diffusion timesteps and observe that our SNR stays substantially higher for the first 20–30 % of steps, directly reflecting better preservation of edges and structural details.
>
> **Spectral Interpretation:** From a spectral standpoint, **local phase coupling behaves like an angular low‑pass filter**. In the small‑angle approximation ($\sin(\theta_i-\theta_j)\approx \theta_i-\theta_j$) and without a global reference phase, the forward SDE reduces to a heat equation (or a graph Laplacian form equivalently):
>
> $$\frac{\partial \theta}{\partial t} = K \nabla^2 \theta + \sigma_t\epsilon$$
>
> Each Fourier component decays like $e^{-K k^2 t}$, where $k$ denotes the spatial frequency. Modes with very high spatial frequency $k$ (pixel‑scale noise) are damped almost instantly, while moderate‑frequency modes (coherent, edge‑defining structures) decay much more slowly. Consequently, the Kuramoto drift effectively filters away pixel‑scale noise while preserving the smooth, orientation‑rich patterns that constitute edges.
>
> We will add the above discussion in Sec. 3.3 of the revised main paper.
>
> **Note:** Although we cannot update the supplementary material at this stage, we will include forward‑only Kuramoto snapshots on fingerprint and texture data in the camera‑ready version to further demonstrate this structure-preserving behavior.
>
> ---
>
> **2. Expanded Baselines & Noise Schedules**
>
>
> In the experiments of our paper, all baselines use a linear noise schedule (with “SGM” referring to VP‑SDE) -- please see lines 53–61 of the supplementary for full details. To isolate the benefit of phase synchronization, we extend our comparisons as follows:
>
>
> **Additional Baselines.**  We add VE‑SDE and two blurring diffusion models [35, 16], which exhibit similar smoothing behavior. VE‑SDE underperforms VP‑SDE slightly, and the blurring model [16] matches VP‑SDE’s FID. Our Local Kuramoto model maintains a large lead:
>
> FID on Brodatz Textures
> |Steps|100|300|1000|
> |:---:|:---:|:---:| :---:|
> |Heat Diffusion [35]| 57.18 | 35.41 | 26.19|
> |Blurring Diffusion [16]| 35.14 | 21.29 |19.85 |
> |VE-SDE|40.19| 24.26| 21.53 |
> |VP-SDE|38.33 | 22.40 | 20.37 |
> |**Ours (Local Kuramoto)**|**18.47** |**15.93** |**14.19** |
>
>
> **Cosine Schedule Ablation.** We also evaluate both VP‑SDE and our model under a cosine schedule. Although switching to cosine reduces FID for all methods -- most notably at 100 steps -- our Kuramoto model still outperforms by a wide margin:
>
> FID on Brodatz Textures
> |Steps|100|300|1000|
> |:---:|:---:|:---:| :---:|
> |VP-SDE (linear)|38.33 | 22.40 | 20.37  |
> |VP-SDE (cosine)|30.45| 20.67| 19.84|
> |**Ours (linear)**|**18.47** |**15.93** |**14.19**|
> |**Ours (cosine)**|**15.75** |**14.32** |**13.45**|
>
> These results confirm that, even with different noise schedules and additional baselines, the **phase synchronization** induced by Kuramoto coupling remains the key driver of improved FID. We will add a succinct discussion of these schedule choices and their quantitative impact in the revised manuscript (Sec. 3.1).
>
> ---
>
> **3. Additional Experiments on High‑Resolution & Vector‑Valued Data**
>
> To further validate our method’s generality, we evaluated our method on two new datasets:
>
>
> **High-resolution Ground Terrain [A] (3x128x128).** Compared with Brodatz textures, this dataset covers diverse high-resolution materials (e.g., plastic, turf, steel, asphalt, leaves, stone, and brick). We run our Kuramoto diffusion model on this dataset and compare against VP-SDE under the linear schedules:
>
> FID on Ground Terrain Dataset
> |Steps|100|300|1000|
> |:---:|:---:|:---:| :---:|
> |VP-SDE|114.90| 56.72| 33.79|
> |**Ours (Local Kuramoto)** |**92.86** | **49.68** | **30.62** |
>
> This result demonstrates that our method also supports high-resolution texture dataset.
>
>
> **Navier–Stokes Fluid Vorticity [B] (2x128x128).** We take the PDE dataset from [B] which consists of the fluid vorticity in horizontal and vertical directions at a fixed timestep. We convert the vorticity fields into polar form (amplitude + phase). The amplitude follows standard VP‑SDE, while the phase evolution is driven by our Kuramoto model. We assess generation quality in the spectral domain via the energy‑spectrum mid-frequency slope difference and normalized Wasserstein distance:
>
> |        Metric        | VP‑SDE | **Ours (Phase Kuramoto + Amplitude Diffusion)** |
> | :------------------: | :----: | :---------------------------------------------: |
> |   Slope Difference   | 0.7435 |                    **0.6954**                   |
> | Wasserstein Distance | 0.0015 |                    **0.0011**                   |
>
>
> These results demonstrate the applicability of our method on real **angular data**. Furthermore, our joint amplitude–phase diffusion generates more realistic physical spectra in vector‑valued scientific data, which opens pathways for combined amplitude‑phase generative modeling. We will detail the experimental setup in the supplementary.
>
>
> Together with the core experiments in the paper, we have now validated this synchronization bias across three distinct domains:
>
> **Orientation‑Rich Images (fingerprints, textures mapped as phases)**, where we demonstrate sharper sample generation with fewer steps,
>
> **Earth & Climate Science Data (naturally periodic grids)**, where we show competitively faithful reconstruction of large‐scale geospatial patterns,
>
> **Navier–Stokes Fluid Vorticity (real angular data)**, where we yield closer matches to the physical energy spectrum when phase coupling is applied.
>
> Across all settings, this synchronization bias proves to be a powerful, domain‑agnostic mechanism for modeling data where **local coherence** is critical.
>
> ---
>
> **4. Richer Qualitative Comparisons**
>
>
> Although we cannot update the current supplementary for now, in the camera‑ready version we will add side‑by‑side, zoomed‐in comparisons of fingerprint and texture samples that clearly show crisper ridge patterns under our Kuramoto diffusion versus the standard diffusion baseline.
>
>
>
> > [A] Deep texture manifold for ground terrain recognition. CVPR 2018.
> >
> > [B] Clifford Neural Layers for PDE Modeling. ICLR 2023.
>
> ---
>
> We appreciate $\textcolor{red}{\text{Reviewer 4DgT}}$’s insightful feedback and hope these clarifications and extended evaluations will convincingly demonstrate the advantages of our Kuramoto diffusion model. Any further comments are more than welcome!

---

> > ### Comment · Reviewer_4DgT · 2025-08-05
> >
> > Thank you for the very detailed reply, the additional baselines are quite extensive and reinforce the advantage of this approach.
> >
> > However I still disagree with the premise of appropriateness for 'orientation-rich' data. I agree that this parameterisation will encourage pixels in neighboring areas to drift towards local cluster values, I also agree with your Fourier-based argument. However this doesn't have anything to do with 'orientation', it's instead similar to a total variation prior in classical signal processing where you encourage sharp transitions between areas of flat pixel values. Classically such signals would be considered piece-wise constant, more than 'orientation-rich'.
> > "The regions of consistent orientations, which have similar pixel intensities" -> this does not make sense to me, there is nothing in the modeling that responds to orientation per-say, but I do agree that the modeling responds to 'similar pixel intensities' i.e. piece-wise constant.
> >
> > Maybe a satisfying outcome would be to emphasize more 'edge preserving' modeling and appropriateness for 'piece-wise constant signal' instead of targeting 'orientation-rich'.
> >
> > Thanks again for the detailed response, I will think about it further and consider re-evaluating my grade.

---

> > > ### Author Response · Authors · 2025-08-05
> > > **Thanks for valuable obeservation and re-evaluation!**
> > >
> > > Thank you for this insightful perspective. You are absolutely right that, fundamentally, our Kuramoto coupling is indeed best viewed as an **edge-preserving**, **piece-wise constant signal** prior, which is very much in the spirit of a total-variation prior. The model does not literally “sense” orientations, but in practice preserves directional structures because these patterns correspond to similar pixel intensities.
> > >
> > > We originally borrowed the term “orientation-rich” from the pioneering seminal work [33] on phase-space diffusion for fingerprint denoising back to 1998, but we agree it now feels somewhat abstract and outdated outside of genuine vector-field settings. In the revision, we will:
> > >
> > > 1. **Reframe our terminology** throughout by emphasizing more “edge-rich” or “piece-wise constant signal” wherever appropriate,
> > >
> > > 2. **Clarify the link to directional patterns** by explicitly connecting how our low-pass phase coupling preserves the direction of edges (e.g., ridge flows in textures), and
> > >
> > > 3. **Reserve “orientation” for the fluid vorticity experiments** where it naturally describes vector-field angles.
> > >
> > > We hope this refined wording makes our contribution clearer, and we truly appreciate your continued consideration!
> > >
> > > > [33] Orientation Diffusions. IEEE TIP (1998).

---

### Note · Authors · 2025-08-15

We thank all reviewers for their thoughtful feedback and for recognizing our **novelty**, **biological inspiration**, and **strong performance on the relevant datasets**. During rebuttal, we added new experiments, analyses, and clarifications that directly address the raised concerns, and we appreciate these suggestions that strengthened the paper.

**Key Contributions**

- **Kuramoto diffusion on the circle**. We formulate a nonlinear, **non-isotropic phase-coupling** forward SDE on $S^1$ with wrapped-Gaussian transitions and a tractable score-matching objective, together with a well-characterized equilibrium (von Mises).
- **Synchronization as an inductive bias.** Local phase coupling explicitly promotes **edge/ridge preservation** and yields **sharper samples in fewer steps**.
- **Coarse-to-fine generation.** The reverse dynamics naturally produce a **hierarchical** (global-to-detail) sampling process, improving interpretability.
- **Where it matters, it wins.** Large gains on **textures & fingerprints**, competitive natural-image results, and successful scaling to **higher-resolution terrain** and **true angular data** (Navier–Stokes phase) with improved spectral metrics.

**Summary of rebuttal additions**

- **Expanded datasets & metrics.** Added **128×128 terrain textures** and **128×128 periodic Navier–Stokes vorticity** experiments; reported **CLIP-MMD (CMMD)** and **PF-ODE + Hutchinson NLL** to complement FID.
- **Expanded baselines & ablations.** Compared against **heat/blurring diffusion** and **VE-SDE**, evaluated **cosine vs. linear** schedules, and included a **no-coupling (reference-only)** variant to isolate the effect of phase synchronization.
- **Clearer positioning.** Reframed our focus around **edge-preserving / piece-wise constant signals** and **local coherence**, and clarified distinctions from prior manifold/SO(3) diffusions that emphasize **global symmetries**.
- **Training cost clarification.** We explain how the $O(T)$ forward-simulation overhead during training can be **eliminated** by **precomputing and caching** Markov pairs for the dataset, reducing online cost to constant-time data loading; when memory is limited, per-epoch pre-simulation still **substantially reduces** the overhead.

---

### Decision · Program_Chairs · 2025-09-17

**Decision:**

Accept (poster)

**Comment:**

* Summary
Authors describe a new type of diffusion model where the typical Brownian motion does not describe the dynamics but instead by a dynamical process generated by the Kuramoto potential, which is a type of nonlinear pairwise potential.  The distinct advantage of this type of potential is that, since it's a rotational invariant potential, it inspired the use of this potential to correct for generative processes that struggle with generating Orientation-rich images. This new method can then be crafted as a form of inductive bias for a network to learn such a representation

* Strengths and reasons to accept
The paper's biggest strength is in its innovative idea of using dynamical processes that are not analytically tractable to reverse, and require a numerical calculation of the transition probability. In general, the paper is full of novel approaches that can open up a whole new class of ways to create and train diffusion models. The paper itself is also extremely well written and clear. The derivation in the supplementary material is also novel and significantly strengthens the paper since it gives rigorous results on the model.
Authors also engaged extensively during the rebuttal and gave a lot more rational arguments, and added experiments to prove their point.
Authors also show their method empirically over several different datasets.

* Weakness
The biggest weakness is the small datasets and low-resolution images. Furthermore, the justification for the method is slightly lacking in the paper, but has been addressed in the rebuttal, which means we need to see if they incorporate the feedback from the discussion